# On the origin of acoustic emission in the stress-induced martensite regime of shape memory alloys

C. Lauhoff [1] ✉, A. Weidner [2], R. Lehnert[2], A. Reul[3], T. Pham [1], M. J. Gutmann [4], P. Krooß[1], W. W. Schmahl[3], H. Biermann [2], H. Seiner [5] & T. Niendorf [1]

The superelastic (SE) deformation behavior of Heusler-type Co-Ni-Ga shape memory alloy (SMA) single crystals is investigated employing complementary in situ techniques. Findings obtained by optical microscopy, neutron diffraction and acoustic emission (AE) provide deep insights into the microstructural events taking place during compressive loading. In addition to the martensitic forward and reverse transformation, unloading of the stress-induced martensite is found to be accompanied by the emission of acoustic signals. In the unloading martensite regime, neutron diffraction gives clear evidence for a change in the volume fractions of martensite domain variants upon unloading, i.e. reorientation of martensite by the growth of one martensite domain variant at the expense of the other one. Hence, it is shown that the origin of the AE signals occurring in that unloading martensite regime can be ascribed to twin boundary motion.

Over the past decades, SMAs have received significant attention by both industry and academia. Due to large reversible strains and their dissipative potential, this class of smart materials is highly attractive for designing efficient and compact solid-state actuator and damping devices, respectively[1–3]. Their unique functional properties, i.e. shape memory effect and superelasticity, are based on a reversible martensitic phase transformation (MPT), i.e. a diffusionless, solid-state phase transition between a high-symmetry austenitic parent phase and a low-symmetry martensitic product phase[1,2]. The SE effect, which is in focus of the present work, represents a mechanical memory behavior. At temperatures above the austenite finish temperature ($A_f$), where the SMA is in a fully austenitic state, the material deforms due to stress-induced MPT and possible reorientation (detwinning) of twinned martensite. Since the stress-induced martensite is unstable at temperatures above $A_f$, the reverse transformation back to austenite occurs upon unloading and, as a consequence, the deformation is fully recovered[1,2].

The successful incorporation of SMAs into envisaged applications depends on a thorough understanding of the MPT behavior. At this point, in situ characterization techniques are particularly suitable to study the ongoing microstructural processes under load. In the past, AE has been already used for characterizing various microstructural phenomena in plenty of different kind of materials, e.g. ultrafine-grained copper[4], transformation-induced plasticity (TRIP) and twinning-induced plasticity (TWIP) steels[5,6] as well as SMA systems[7,8]. In particular, AE is one of the methods of choice for studying microstructural phenomena in SMAs, since among the plethora of in situ characterization techniques the AE method is one of the scarce real-time methods, providing volume integrated information at high time resolution in the range of microseconds[9]. However, AE is an indirect method and, thus, needs to be corroborated by further characterization techniques, i.e. either by diffraction (neutron, X-ray, and electron) or imaging (optical and electron microscopy) techniques. In addition, further work is still needed to explore its full potential.

[1]Institute of Materials Engineering, Universität Kassel, Kassel, Germany. [2]Institute of Materials Engineering, Technische Universität Bergakademie Freiberg, Freiberg, Germany. [3]Department of Earth and Environmental Sciences, Applied Crystallography, Ludwig-Maximilians-Universität, Munich, Germany. [4]ISIS Facility, Rutherford Appleton Laboratory, Chilton, Didcot, Oxfordshire, United Kingdom. [5]Institute of Thermomechanics, Czech Academy of Sciences, Prague, Czech Republic. ✉e-mail: lauhoff@uni-kassel.de

Planes et al.[10] and Niemann et al.[11] performed in situ AE measurements on Cu-based, Ni-Al and Fe-Pd as well as Ni-Mn-Ga polycrystalline SMAs, respectively. These AE measurements were performed in a threshold-based data acquisition and relied on time-domain analysis of the data such as number of hits, maximum hit amplitude and the absolute energy, which is given by the envelope of a hit. As a result, the threshold-based data acquisition focuses only on transient, burst-type signals and disregards all continuous-type signals. The latter, however, may also contain valuable information, in particular stemming from AE sources operating at lower velocities such as dislocation movements. In the aforementioned studies[10,11], multiple AE sensors were used for the localization of the AE events in combination with optical observations. The authors concluded that the AE measurements are a valuable tool to discover the kinetics of the transformation behavior of SMAs.

In addition, Bonnot et al.[12] performed in situ threshold-based AE measurements on Cu-Zn-Al SMA, showing an asymmetry of the stress-induced MPT during the loading and unloading path. However, as the authors also analyzed the AE data only in the time-domain by calculating the AE activity (number of events per unit time; dN/dt), a clear distinction between different microstructural processes based on the AE sources was not possible[12]. Such asymmetry in the forward and reverse transformation was also found in a previous study detailing the MPT behavior of an Fe-Mn-Al-Ni SMA[7]. In that study, moreover, transient AE signals were already detected during the unloading of stress-induced martensite, i.e. before the stress plateau of the reverse transformation was reached. Accompanied by optical observations, the origin of these signals was found to be the reverse transformation of tiny martensitic areas back into austenite[7].

The aim of the present study is to shed light on continuous-type AE signals, which also appear in the unloading martensite regime of SMAs. The material of interest in this work are Co-Ni-Ga single crystals with ⟨001⟩ crystal orientation. Single crystals were used instead of polycrystalline material to allow for a systematic assessment of the operant microstructural mechanisms. In Co-Ni-Ga alloys, which are characterized by a pronounced anisotropy of the MPT behavior[13,14], polycrystals suffer premature intergranular cracking upon thermomechanical loading due to incompatibilities at high-angle grain boundaries[15–17]. Single crystals with the loading axis along ⟨001⟩, in turn, were selected because of the high slip resistance in this orientation, allowing for excellent transformation recoverability and cyclic functional stability under SE testing conditions[14,18–20]. As a result of the active ⟨001⟩{110} slip system in the B2-ordered austenite, the choice of the ⟨001⟩ orientation minimizes the effect of dislocation activities on the deformation behavior[21]. Detailed in situ analysis, i.e. AE as well as optical microscopy (OM) and neutron diffraction, were carried out to analyze the deformation behavior in detail during compression SE experiments. The results obtained with these complementary techniques reveal the deformation mechanisms operant during SE testing and clearly unfold the origin of the AE signals. Twin boundary motion causing the growth of one martensite domain variant at the expense of the other one, known in the literature as martensite reorientation (detwinning/re-twinning), is detected for the unloading martensite regime.

## Results

### In situ optical microscopy

Figure 1 shows the SE response of Co-Ni-Ga under compression at 100 °C and the corresponding optical micrographs recorded during loading (a-e) and unloading (f-h), revealing the morphology of the stress-induced martensite. As can be deduced from the stress-strain curve, a SE behavior with full strain recovery is present for the ⟨001⟩-oriented single-crystalline material in the present study. Both the lack of available slip systems[21] as well as the favorable crystallographic orientation for the MPT (cf. resolved shear stress factor criterion in ref.

13) associated with easy motion of the martensitic interfaces are reasons for this excellent reversibility.

Heusler-type Co-Ni-Ga SMAs undergo a thermoelastic MPT from cubic B2-ordered austenite to non-modulated tetragonal martensite with L1$_0$ structure[22,23]. As already shown in a previous study[24], the stress-induced forward transformation in the solution-annealed (precipitate-free) condition under compressive loading is characterized by the nucleation and growth of a dominant martensite plate (bottom right corner in Fig. 1b-d). The martensite plates/laminates can be distinguished from the austenitic parent phase by the optical contrast. Due to their direct and well-defined interface with the austenitic matrix, called habit plane, such plates are termed habit plane variants (HPVs). Here, the habit planes are seen on the (010) lateral surface (Fig. 1). According to the calculation presented in the *Supplementary Material*, the habit planes are aligned to the near-(101) plane of the austenite, which is in good agreement with their traces being inclined by around 45° to the loading (compression) axis. When ⟨001⟩-oriented Co-Ni-Ga is subjected to compressive load, furthermore, the martensite within the HPVs comprises two twin-related domain variants[13,18,25], called correspondent variant pair (CVP, s. following paragraph for more details). While Niklasch et al.[25] were able to visualize the twinned martensitic microstructure by means of high-resolution magnetic force microscopy (MFM), the OM setup used in the present study does not allow to observe the martensite domain variants. Hence, these domain variants, which are separated by twin boundaries, will be assessed hereafter based on neutron diffraction.

Various models, e.g. Bain, Kurdjumow-Sachs, Nishiyama-Wassermann, Pitsch, and Greninger-Troiano, have been introduced in the past to describe the lattice relationships between the austenitic parent and martensitic product phase in different alloy systems. In case of Heusler-type Co-Ni-Ga, the Bain model allows to specify the martensite variant selection under uniaxial loading conditions in a simplified but precise manner. As schematically illustrated in Fig. 2, in theory, a total of three different tetragonal domains (Bain-correspondent variants, BCV) can be formed during the cubic-to-tetragonal phase transition. The extensional c-axes of these BCVs are parallel to the main cubic axis of the parent B2-austenite. However, upon compressive loading along the ⟨001⟩ crystal direction, i.e. the loading condition in the present study, the formation of the martensite domain with its c-axis parallel to the loading direction, namely BCV$_3$, is energetically suppressed. Only the two domains with their c-axes perpendicular to the loading direction, i.e. BCV$_1$ and BCV$_2$ (Fig. 2), can occur and eventually form CVP systems, where BCV$_1$ is assumed to be the variant with the c-axis aligned with the [100] axis in austenite, and BCV$_2$ is the one with the c-axis aligned with [010]. Twin planes are of type {101} in the tetragonal lattice in martensite, which (for the two considered variants) are lattice-correspondent planes to (110) and (1$\bar{1}$0) in austenite, respectively. In other words, two possible twin configurations exist along (110)$_{B2}$ and (1$\bar{1}$0)$_{B2}$ under uniaxial compression along ⟨001⟩[18]. The formation of a coherent and stress-free twin plane within a CVP, in turn, necessitates a rotation of the final martensite domain variants (cf. domain variants V$_1$ and V$_2$ introduced in the next section) in relation to the original orientation of the BCVs, where the rotation angle φ depends on the lattice parameters a and c.

Upon unloading from −5%, where the MPT is fully accomplished (Fig. 1e), the stress-induced martensite is thermodynamically unstable and the reverse transformation to austenite sets in (Fig. 1f-h). In first approximation, all habit planes that formed during forward and reverse transformation can be considered as planes parallel to each other. However, as shown by a calculation in the *Supplementary Material*, there are up to eight habit plane orientations possible between austenite and the mixture of BCV$_1$ and BCV$_2$, indicating that the real microstructure could be much richer. And indeed, on closer inspection of the OM observations, a slight splitting of some habit plane traces (cf. habit plane traces in the upper left corner of Fig. 2c, d,

**Fig. 1 | Characteristic superelastic (SE) stress-strain hysteresis under compressive loading at 100 °C and in situ optical microscopy (OM) results of an ⟨001⟩-oriented Co-Ni-Ga single crystal in solution-annealed condition.** The micrographs were recorded during the single cycle test upon loading and unloading at strain values marked by the red points (**a**–**h**). Loading direction (LD) is horizontal as marked in the upper right corner of subimage (**a**). Source data of the stress-strain curve are provided as a Source Data file. See main text for details.

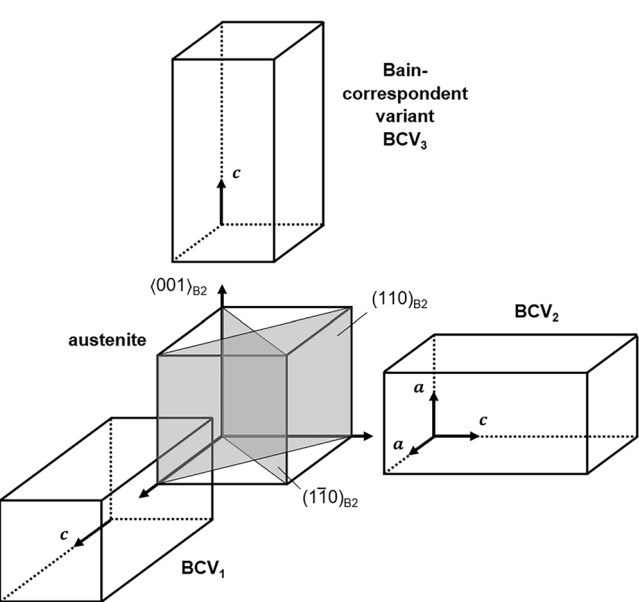

**Fig. 2 | Schematic illustrating the Bain orientation relationship of tetragonal martensite domain variant orientations to cubic austenite.** *a* and *c* are the lattice parameters of the tetragonal martensite. Twin planes are of type {110}.

and g, for example) as well as a slight rotation of the habit plane trace orientation with increasing strain (cf. Figure 2b and c, for example) can be seen in the optical micrographs. This first phenomenon is a shadow effect, which is caused by the characteristic surface relief of a material undergoing a MPT[2]. The latter, in turn, is seen to be based on incompatibilities and further accommodation processes between the austenitic parent phase and the twinned martensite. As discussed below and detailed in the *Supplementary Material*, the domain variant selection in the stress-induced martensite is supposed to be affected by a competition between strain compatibility at the habit planes and boundary conditions related to the grips. The latter is thought to lead to the observed irregularities in the habit plane appearance. Despite these irregularities, however, all HPV plates/laminates formed during testing (Fig. 1) consist of the same two domain variants of martensite, i.e. the same CVP system, as shown by the neutron diffraction analysis hereafter. This finding, in turn, is in excellent agreement with in situ data on ⟨001⟩-oriented Co-Ni-Ga single crystals in solution-annealed condition already tested under compression previously[18,24].

## In situ neutron diffraction

As mentioned before, a deformation related to a twin reorientation mechanism within a CVP system, i.e. growth and shrinkage of one martensite domain variant at the expense of the other one during loading and unloading, respectively, cannot be assessed via the in situ

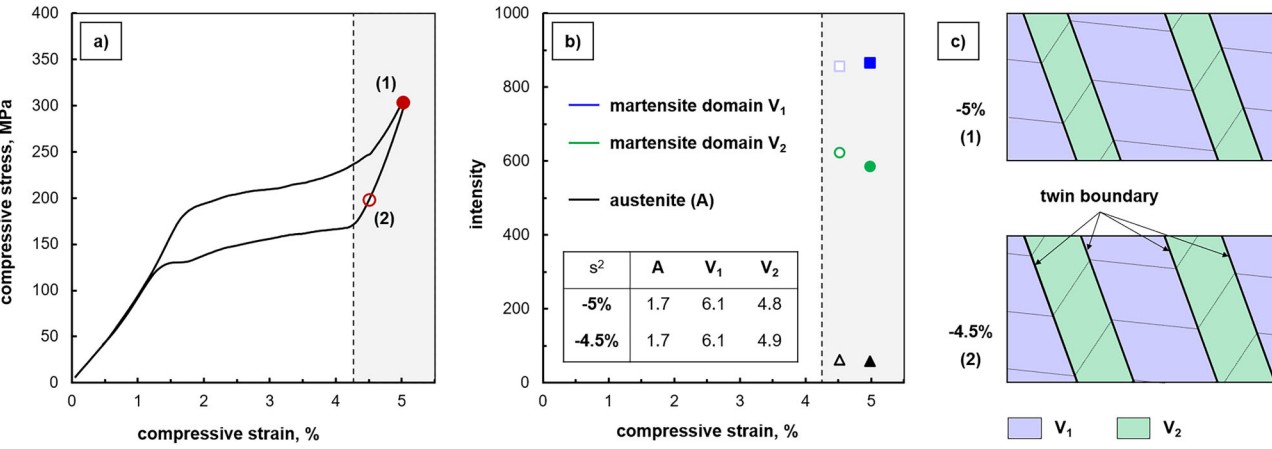

**Fig. 3 | In situ neutron diffraction results obtained during a superelastic (SE) compression cycle at 100 °C for an ⟨001⟩-oriented Co-Ni-Ga single crystal in solution-annealed condition. a** Characteristic SE stress-strain curve. **b** Diffraction intensities obtained for austenite (black triangles) as well as martensite domain variant $V_1$ (blue squares) and $V_2$ (green circles) at −5% (filled symbols) and −4.5% (open symbols), respectively. **c** Schematic illustrating the re-twinning process in the martensite regime (grey region) upon unloading. The specific stress-strain stages investigated are highlighted by the red point (−5%) and circle (−4.5%) in (**a**). The inset in **b** reveals the intensity variance ($s^2$) of the analyzed diffraction data. Source data are provided as a Source Data file. See main text for details.

optical analysis in Fig. 1. Therefore, in situ time-of-flight (TOF) neutron diffraction experiments have been additionally conducted in the martensite regime upon unloading (s. Fig. 3a) in order to shed light on the martensite domain variant selection and the volume ratio of the individual domains. The unloading martensite regime was selected for the neutron diffraction experiments due to the following two reasons: (1) Superimposed effects by a simultaneous reverse transformation of martensite into austenite were not to be expected and, thus, did not affect the evaluation of the domain (twin) reorientation mechanism. (2) In comparison to the loading regime of martensite (s. segment 4 in Fig. 5b), the intensity of the AE signals was found to be unexpectedly higher (s. segment 5 in Fig. 5b), indicating more pronounced twin reorientation activities in this region (s. discussions in the remainder of the text). In the past, neutron diffraction was demonstrated to be a valuable method for an in-depth phase analysis, providing structural information of bulk samples[18,26,27]. As shown in previous studies for various SMA systems, phase (volume) fractions of martensite domain variants and, thus, elementary deformation mechanisms like twinning under mechanical loading could be assessed in polycrystalline material[28] as well as single crystals[29–31].

The integrated diffraction peak intensities calculated from data recorded at −5% and −4.5% compressive strain are depicted in Fig. 3b. In general, such diffraction peak intensities are aligned with the volume fraction of individual phases. Diffraction data at both stress-strain stages (cf. marks in Fig. 3a) revealed the presence of two martensite domain variants labeled as $V_1$ and $V_2$ in the remainder of the text, respectively. It is important to note that intensities related to additional domain variants were not detected. Only some minor intensities from the austenitic parent phase were found, indicating that the stress-induced MPT was only partially completed in the present SE cycle and untransformed regions remained somewhere in the microstructure after loading to the maximum stress level at −5% compressive strain. However, similar findings reported elsewhere[32,33] indicate that this holds true in other SMAs as well. While the austenite has been indexed as a cubic structure with space group Pm-3m (#221), the stress-induced martensite domains feature a body-centered tetragonal (bct) structure with space group P4/mmm (#123), which is fully in line with refs. 18,23,27. According to the Bain model and the lattice correspondence presented in Fig. 2 and the discussion detailed in the *Supplementary Material*, the two martensite domains ($V_1$ and $V_2$) form a CVP. The detection of only these two domain variants indicates that all stress-induced martensite plates formed in the sample comprise the

same CVP system, with a single orientation of twin planes between these two domain variants. In contrary, in a mixture of different CVP systems connected through macro-twin interfaces, there would be small variations of martensite lattice rotations, leading to a splitting of the diffraction peaks in the neutron diffraction experiment, which however, was not observed in the present study as mentioned before.

The analysis of the integrated diffraction peak intensities allows to quantitatively evaluate the actual twinning state. From the results depicted in Fig. 3b, it is obvious that at maximum applied compressive strain (−5%) the intensity of domain variant $V_1$ (blue) is increased by about 50% compared to domain variant $V_2$ (green), i.e. $V_1$ is the volume-dominant domain within the stress-induced martensite. Upon unloading from −5% to −4.5% compressive strain, however, a slight increase in the intensity of $V_2$ and, concomitantly, an intensity decrease of $V_1$ can be seen. In other words, the intensity ratio $V_2/V_1$ changes from 0.68 (−5%) to 0.73 (−4.5%), revealing broadening of the $V_2$ twin bands in the CVP laminate via twin boundary motion as schematically illustrated in Fig. 3c. At this point, it is important to point towards the low variance of the intensity data points (s. inset of Fig. 3b), indicating that the results are adequate for reliable argumentation on twinning mechanisms and, thus, the origin of AE signals as detailed hereafter. Furthermore, please note that the stress-strain stage at −4.5%, where the diffraction measurement was conducted, is clearly in the martensite regime, i.e. above the critical stress for the onset of the reverse transformation into austenite. Hence, the decrease in the intensity of $V_1$ cannot be attributed to a significant growth of austenite on expense of this variant, although the presence of some residual austenite is confirmed by the measurement, and the intensity of the related peak indeed slightly increases with unloading.

The intensity (volume) ratio $V_2/V_1$ in the compressed crystal can be interpreted as a result of energy minimization under the given external load and prescribed boundary conditions[34]. For the prescribed axial contraction and with considered rigid boundary conditions at the grips of the loading device, the energy minimizer is the 1:1 mixture of martensite domain variants. In contrast, the neutron diffraction data show that one variant is dominant in the mixture. This can be rationalized through the mechanism how the stress-induced MPT proceeds (s. the *Supplementary Material* for more details and explicit calculations): at the habit plane, the strain compatibility conditions require a specific $V_2/V_1$ ratio that is close to 1:2. Thus, the HPV plates/laminates nucleate with this 1:2 $V_2/V_1$ ratio and grow towards the grips,

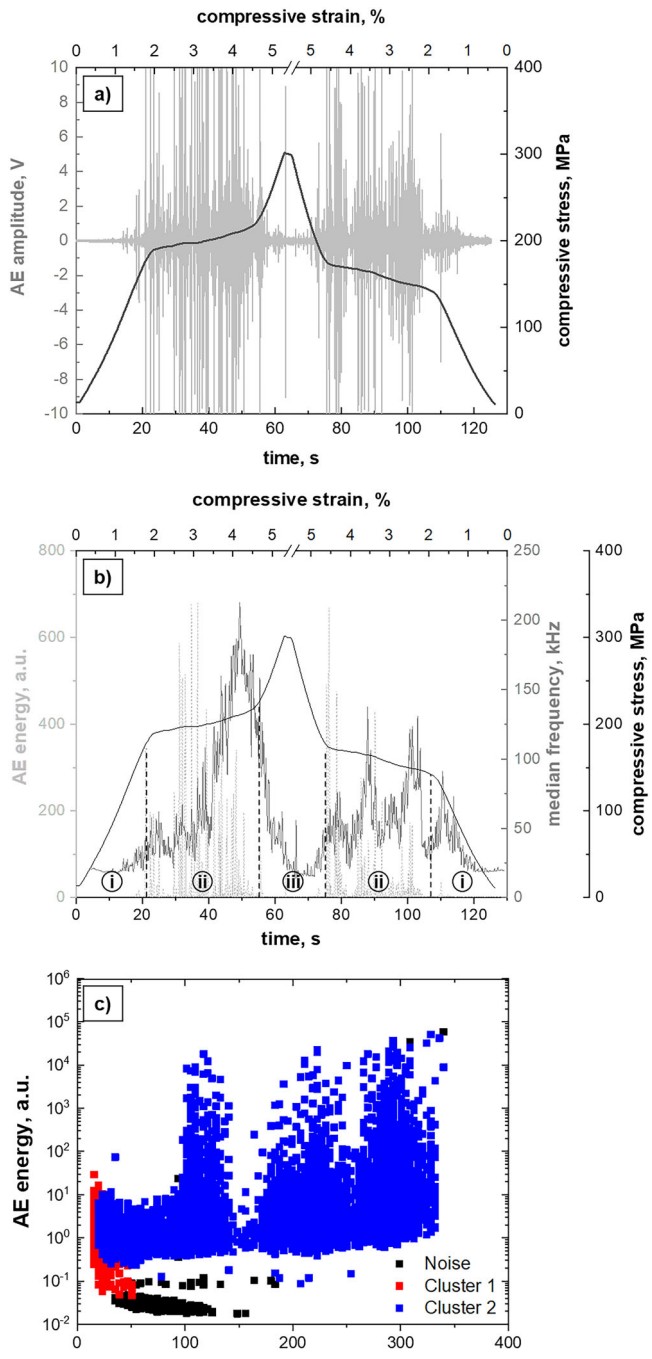

**Fig. 4 | In situ acoustic emission (AE) results obtained during a superelastic (SE) compression cycle at 100 °C for an ⟨001⟩-oriented Co-Ni-Ga single crystal in solution-annealed condition. a** AE data stream (grey) plotted in combination with stress (black) vs. time and strain, respectively. **b** Evolution of the two main parameters of noise-corrected and normalized power spectral density functions (PSDFs), namely AE energy $E$ (dashed grey, positive part), and median frequency $f_m$ (dark grey) in combination with stress (black) vs. time and strain, respectively. **c** Bivariate scatter plot of AE energy $E$ vs. median frequency $f_m$, revealing three individual clusters as a result of the adaptive sequential k-means (ASK) clustering algorithm. Source data are provided as a Source Data file. See main text for details.

where the 1:1 mixture is preferred instead. The resulting ratio close to 0.7 (~1:1.5) is then a compromise between these two competing effects, and can be heterogeneously distributed over the crystal and the stress-induced martensite, being closer to 1:1 close to the grips, and closer to 1:2 in the central part of the crystal. Consequently, the crystal

at the end of the loading plateau is elastically stressed not only because of the axial force from the loading device, but also because of accommodating the boundary condition at the grips and compensating the variations of the $V_2/V_1$ ratio along the sample. As a result, any changes of the boundary conditions, such as unloading the sample from −5% to −4.5% compressive strain, may result in further rearrangement of the existing two domain variants.

The change in the $V_2/V_1$ ratio with unloading from −5% to −4.5% compressive strain also explains the difference between the habit plane microstructures observed during the forward and reverse MPT. Because the ratio evolves more towards 1:1 (from 0.68 (−5%) to 0.73 (−4.5%)), and plausibly reaches even higher values until the reverse plateau is reached), it is impossible for the microstructure to form directly a single austenite band connected to martensite via two habit planes, as these would require the 1:2 ratio in its vicinity. Instead, a finer microstructure needs to form, in which, for example, a HPV plate/laminate with the CVP system is encapsulated between thin austenite bands and, thus, can attain the 1:2 ratio without interacting with the microstructure close to the grips. The variations in the shadowing contrast and other irregularities seen in the micrographs in Fig. 1f and g strongly support such considerations.

### Acoustic emission

Complementary to the in situ OM analysis (Fig. 1) and corroborating the in situ neutron diffraction experiment (Fig. 3), the MPT behavior has been systematically investigated using in situ AE measurements during a SE compression test. The stress-strain response of the Co-Ni-Ga single crystal probed is similar to those shown in Figs. 1 and 3. Figure 4a shows the compressive stress response vs. time and strain in combination with the threshold-less recorded AE data stream (amplitude). It is obvious that immediately with passing the critical stress for the onset of the stress-induced MPT at around 200 MPa a large number of transient signals with huge amplitudes (even crossing the detection limit of 10 V of the transducer) occur in the AE data stream and are present over the entire forward transformation plateau. However, first transient signals with lower amplitudes already occurred right before the critical transformation stress is reached. At the end of the transformation plateau (at around 55 s and about −4.5% applied strain), when the stress starts to increase, the number and amplitude of the transient signals significantly decrease. A similar behavior is observed during the unloading path in opposite direction: (a) Small amplitude signals in the unloading regime of the martensite between 300 MPa and 200 MPa compressive stress, and (b) huge amplitude signals in the plateau region of the reverse transformation into austenite between −4.5% and −1.5% compressive strain are present. However, the complete AE data stream in the time domain (Fig. 4a) does not allow for any conclusions on the origins of the acoustic signals. Therefore, Fast Fourier transformation (FFT) and noise correction were applied to the AE data stream. Further analysis was done in the frequency domain on the basis of power spectral density functions (PSDFs) and their two fundamental parameters, i.e. AE energy $E$ and median frequency $f_m$.

The evolution of these two parameters over the entire SE cycle is shown in Fig. 4b, illustrating the positive part of the AE energy $E$ (dashed light grey), the median frequency $f_m$ (dark grey), and the compressive stress (black) as a function of time and compressive strain, respectively. It turns out that the noise-corrected AE energy $E$ is relatively high in the two plateau regions (forward and reverse transformation) labeled with (ii) in Fig. 4b. However, it is very low (but not zero) in the regions of elastic loading and unloading of austenite (regions labelled with (i) in Fig. 4b). Moreover, also in the regions labeled with (iii), i.e. between −4.5% compressive strain up to the maximum compressive stress at −5% (loading path) and back to −4.5% (unloading path), the AE energy $E$ is very low, i.e. on a level found in the

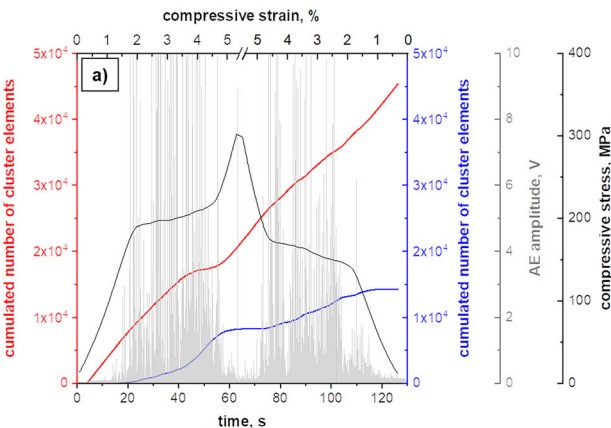
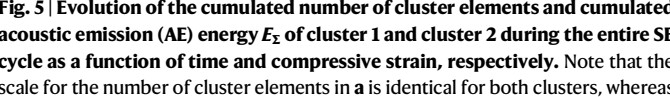
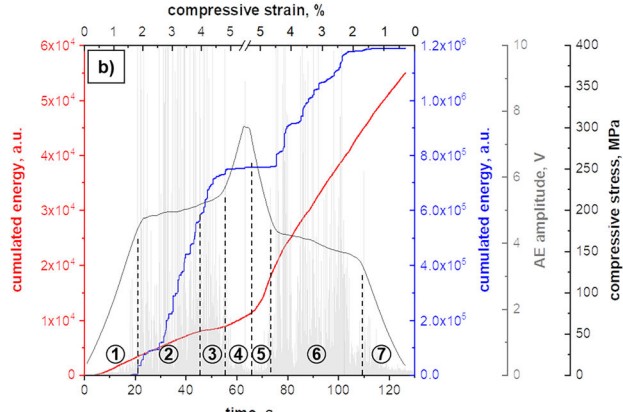

**Fig. 5 | Evolution of the cumulated number of cluster elements and cumulated acoustic emission (AE) energy $E_\Sigma$ of cluster 1 and cluster 2 during the entire SE cycle as a function of time and compressive strain, respectively.** Note that the scale for the number of cluster elements in **a** is identical for both clusters, whereas the scale for the cumulated AE energy $E_\Sigma$ in **b** is significantly different for cluster 1 (red; max. $6 \times 10^4$) and cluster 2 (blue; max. $1.2 \times 10^6$), respectively. Source data are provided as a Source Data file. See main text for details.

elastic loading and unloading path of the austenite (cf. regions (i) in Fig. 4b).

The median frequency $f_m$, in turn, demonstrates a similar trend during the SE cycle. A pronounced increase can be seen from passing the critical stress for the onset of the stress-induced MPT until the end of the transformation plateau. This increase in median frequency $f_m$ is caused by the changes in the microstructure, i.e. a reduction of the mean free path/volume for the ongoing processes (s. discussions in the remainder of the text). Then, the median frequency $f_m$ drops down continuously up to the point where the maximum strain level of −5% is reached, and remains at a low level during the subsequent unloading in the martensite regime until the plateau region of the reverse transformation occurs.

Taking a closer look at the parameters $E$ and $f_m$ and performing a cluster analysis according to the adaptive sequential k-means (ASK) algorithm developed by Pomponi and Vinogradov[35], at least three different clusters of AE signals can be separated. Figure 4c depicts the bi-variate scatter plot of AE energy $E$ vs. median frequency $f_m$. Beside a cluster of noise (black symbols) with very low energies and a wider scatter of median frequencies, two other clusters (red and blue) with distinct different parameters $E$ and $f_m$ were identified. Cluster 1 (Cl 1; red symbols) is characterized by a quite narrow band of median frequency $f_m$ of about 20 to 60 kHz. The AE energies $E$ of this cluster are relatively low, however, still above of those of the noise signals. It is well known from literature and previous studies of some of the authors that low energy and low median frequency AE signals are related to microstructural processes occurring with lower velocities such as dislocation movement or the movement of interfaces[6]. In contrast, cluster 2 (Cl 2; blue signals) contains signals with significantly higher AE energies $E$, which are also spread over a wider range of median frequencies $f_m$ from 20 up to 350 kHz. This kind of signals is known to be related to fast processes occurring with the velocity of sound such as brittle crack formation, MPT, mechanical twinning or even detwinning in hexagonal close-packed (hcp) materials such as magnesium[6,36,37]. Furthermore, it seems that Cl 2 consists of three individual subclusters (Fig. 4c) with centroids of the median frequencies $f_m$ around 100 kHz, 200 kHz and 300 kHz, respectively. However, the underlying Kullback–Leibeler divergence criterion of the applied adaptive sequential k-means algorithm was not able to separate Cl 2 into three subclusters. As will be detailed in the remainder of the text, all signals of Cl 2 are caused by the MPT, while the specific AE energy $E$ and median frequencies $f_m$ values are related to the dimensions of the martensitic microstructure, i.e. the material volume involved in the MPT and the length of the habit

planes, respectively. In turn, the missing high-amplitude, high-energy signals (Fig. 4a) as well as the continuous drop in both the AE energy $E$ as well as the median frequency $f_m$ (Fig. 4b) are clear indicators that no forward or reverse MPT occurs beyond the plateau regions of the stress-strain hysteresis loop.

For an in-depth analysis of Cl 1 (red) and Cl 2 (blue), Fig. 5 shows the cumulated number of cluster elements (a) as well as the cumulated AE energy $E_\Sigma$ (b), allowing to follow their evolution over the entire SE cycle. The main findings are as follows:

(1) Cl 1 starts to appear at around 5 s, i.e. before the critical transformation stress for the forward MPT is reached, and evolves over the entire SE cycle, i.e. even beyond the stress plateaus of the forward and reverse MPT. Therefore, Cl 1 is not related to the forward and reverse MPT.

(2) The cumulated number of cluster elements of Cl 1 is continuously increasing. As the related AE energy $E$ value per individual cluster element is very small, a smooth progression of the cumulated AE energy $E_\Sigma$ is seen. However, both curves, i.e. for the cumulated number of cluster elements as well as the cumulated AE energy $E_\Sigma$ feature different slopes with time and strain, respectively (cf. segments 1–7 in Fig. 5b).

(3) Cl 2 starts to evolve with reaching the critical transformation stress for the forward MPT (at around 20 s). Noteworthy, the curves for the cumulated number of cluster elements as well as the related cumulated AE energy $E_\Sigma$ continuously increase in the regions of the two stress plateaus related to the forward and reverse MPT. In the other regions, in turn, both curves are leveled out, indicating that Cl 2 is solely related to the MPT.

(4) While the cumulated number of cluster elements of Cl 2 is low compared to Cl 1, the individual AE energy $E$ values of each cluster element are significantly higher, resulting in a stair-case shaped evolution of the cumulated AE energy $E_\Sigma$ of Cl 2.

(5) Noteworthy, Cl 2 reveals different total amounts of released cumulated AE energy $E_\Sigma$ between the forward and reverse MPT (cf. blue curve in segments 2 and 6 of Fig. 5b, respectively). This asymmetry can be directly deduced from the macroscopic optical observations in Fig. 1. The released AE energy of a transformation event depends on the dimensions of the martensite/austenite plates and the involved volume fraction, i.e. the larger the volume fraction the higher the energy release. Since the austenitic bands formed from the stress-induced martensite upon unloading are smaller in their dimensions (volume) compared to the dominant martensite plate evolving

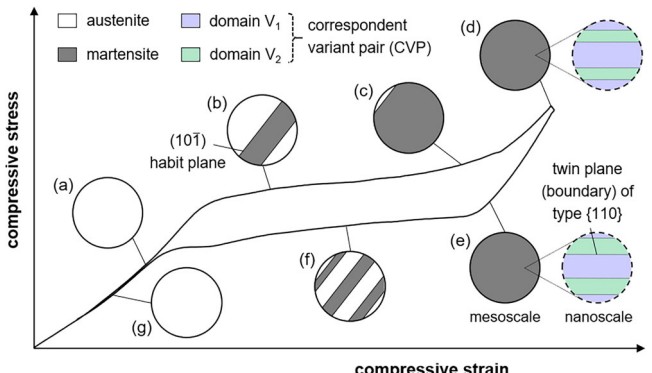

**Fig. 6 | Schematic illustrating the relevant deformation mechanisms in ⟨001⟩-oriented Co-Ni-Ga SMA single crystals in solution-annealed condition under superelastic (SE) compressive loading at 100 °C.** The stress-strain hysteresis is recompiled from the neutron diffraction experiment in Fig. 3a. The solid and dotted circles depict microstructural features on the meso- and nanoscopic scale, respectively. See legend and main text for details.

during the loading path, the cumulated AE energy $E_\Sigma$ of the signals related to the reverse transformation is less compared to the forward transformation even though the total transforming volume during forward and reverse MPT is the same (s. fully reversible SE stress-strain response in Fig. 1). This asymmetry was already described for ⟨001⟩-oriented Fe-Mn-Al-Ni SMA in ref. 7.

While the comparison of the SE stress-strain response with the evolution of the cumulated number of cluster elements (Fig. 5a) and the cumulated AE energy $E_\Sigma$ (Fig. 5b) already allows for first interpretations of the AE signals, e.g. that Cl 2 is solely related to the MPT as stated above, other phenomena require further analysis by considering the corroborating in situ findings from the OM and neutron diffraction measurements. In particular, a distinct increase in the cumulated AE energy $E_\Sigma$ of Cl 1 is visible in the martensite regime (s. segments 4 and 5 in Fig. 5b). The main objective of the present study is to shed light on the origin of these AE signals upon loading and unloading of stress-induced martensite in the Co-Ni-Ga shape memory alloy.

## Discussion

In the present study, the deformation behavior of Heusler-type Co-Ni-Ga SMA single crystals has been assessed under SE compressive loading conditions employing various complementary in situ techniques. Beside the OM observations showing the stress-induced forward and reverse MPT accompanied by the movement of the interfaces between austenite and martensite (habit planes, cf. Fig. 1), the neutron diffraction data obtained in the unloading martensite regime have clearly revealed a martensite reorientation (twinning) mechanism, i.e. the growth of the minor dominant domain variant ($V_2$) at the expense of the major one ($V_1$) (cf. Figure 3c) by the movement of twin boundaries. The current investigations aim at the correlation between these microstructural processes and the AE data (Figs. 5 and 6) recorded during the entire SE cycle. While Cl 2 is already well-known to be governed by the MPT, Cl 1 is related to several sources all causing low-energy, low median frequency signals, i.e. (i) dislocation movement (if any), (ii) movement of austenite/martensite interphase boundaries (habit planes) and (iii) movement of twin boundaries as a result of twin reorientation processes. Hence, the discussions presented in the following will focus on the interpretation of the changes in the cumulated AE energy $E_\Sigma$ of Cl 1 (Fig. 5b), allowing to shed light on the motion of interphase and/or twin boundaries during the SE behavior of SMAs.

Beforehand, two facts have to be mentioned: (i) Under the present loading conditions, i.e. compression along ⟨001⟩, twinning of martensite, i.e. CVP formation within an HPV plate/laminate, occurs simultaneously with the stress-induced formation of the martensite plate at the velocity of sound[24], and (ii) the process of martensite domain reorientation, i.e. detwinning and re-twinning, in SMAs is different to mechanical detwinning in hcp materials such as Ti- or Mg-alloys, where complete mechanical twins disappear spontaneously with the velocity of sound[36–38]. In contrast to hcp materials, the twinning mechanisms in SMAs are related to the movement of twin boundaries between the two martensitic domain variants, forming one CVP[24,39]. Moreover, the movement of these twin boundaries during martensite reorientation (detwinning and re-twinning) is expected to occur at low velocity similar to dislocation-mediated processes. In addition, the same holds true for the movement of the interphase boundaries (habit planes) between austenite and martensite.

The cluster analysis of the AE signals (Fig. 4c) and their evolution over the entire SE cycle (Fig. 5) reveal that Cl 1 appears already prior to the onset of the MPT, which is indicated by Cl 2 starting exactly when reaching the critical stress for the MPT. However, while Cl 2 nearly levels out at the end of the stress plateau region and remains more or less constant until the critical stress for the reverse transformation is reached upon unloading, where again a significant increase in Cl 2 is visible, Cl 1 continuously evolves with even different intensities (slopes) in various regions over the entire loading-unloading cycle (cf. individual segments 1–7 in Fig. 5b). In particular in the region, where Cl 2 is on a relatively constant level since no significant MPT processes (forward as well as reverse transformation) occur, significant changes in the cumulated AE energy $E_\Sigma$ of Cl 1 can be observed.

As mentioned above, Cl 1 stems from microstructural processes, which are occurring at lower velocity. According to this, the evolution of its cumulated AE energy $E_\Sigma$ slope in the different segments over the entire SE cycle can be explained as follows: Cl 1 starts to evolve already in the region of the elastic loading of austenite, i.e. directly at the beginning of the SE cycle (cf. segment 1 at around 5 s). While movements of austenite/martensite interfaces (habit planes) and twin boundaries can be safely excluded here, the influence of dislocation slip is also seen to be neglectable. In alloys with B2 structure, dislocation slip occurs along the ⟨001⟩ direction on {110} planes and, thus, application of stress along ⟨001⟩ leads to zero Schmidt factor[21]. Hence, the origin of the AE signals in the austenite regime, i.e. upon loading (segment 1) and unloading (segment 7), cannot be finally assessed based on the results presented and needs further clarification in future work.

After its onset in the elastic austenite regime, then, the cumulated AE energy $E_\Sigma$ of Cl 1 is increasing continuously with the same slope until shortly before the end of the forward transformation plateau (s. segment 2). Within this plateau region, stress-induced martensite is formed accompanied by the movement of single austenite/martensite interfaces (habit planes) as clearly seen from the in situ optical micrographs in Fig. 1b-d. Due to the quasi-static test procedure with a nominal strain rate of $1 \times 10^{-3}\,\mathrm{s}^{-1}$ (cf. experimental details in the Methods section), the motion rate (velocity) of these habit planes is relatively slow and, thus, is supposed to contribute to the AE signals belonging to Cl 1, i.e. low AE energy levels $E$ and low median frequencies $f_m$ (cf. Figure 4c). Nearby the end but still within the stress plateau, however, the slope of the cumulated AE energy $E_\Sigma$ of Cl 1 decreases (s. segment 3). This, in turn, can be rationalized by the fully completed stress-induced MPT (from a macroscopically point of view, cf. Fig. 1d). As a result, habit plane motions are supposed to be diminished and, thus, no distinct microstructural events are contributing to AE signals of Cl 1 in this stage (segment 3).

After the forward transformation plateau, once Cl 2 remains more or less constant, Cl 1 shows a significant increase during both further loading in the martensite regime up to the maximum compressive

strain level of −5% (segment 4) as well as unloading of martensite until the onset of the stress plateau of the reverse transformation (segment 5). Here, the movement of twin boundaries between $V_1$ and $V_2$ is supposed to contribute to signals of Cl 1, which can be rationalized based on the findings obtained by the neutron diffraction experiments (Fig. 3). The martensite domain variant $V_1$ has the highest diffraction intensity at the maximum stress. Upon unloading, the intensity ratio $V_2/V_1$ increases from 0.68 (−5%) to 0.73 (−4.5%). This behavior can be interpreted as the growth of $V_2$ at the expense of $V_1$ during unloading, indicating that the microstructure does not simply remain in the state attained during loading or deform solely through the relaxation of elastic strains. Rather, it continues to evolve throughout the unloading process. The twin boundary motion within the HPV plates/laminates comprising the CVP system causes the steady increase in the cumulated AE energy $E_\Sigma$ of Cl 1 in the stage of unloading of the stress-induced martensite. Even though not directly shown in the present study, equally, the contribution of twin boundary motion is assumed to be responsible for the corresponding signals of Cl 1 during the previous loading step up to the maximum stress level at −5% compressive strain. In this region, however, the opposite behavior is expected, i.e. detwinning, where $V_1$ is growing at the expense of $V_2$. This behavior is known from one of the previous studies[23], where detwinning of martensite has been observed in the loading regime, i.e. the martensite domain variant, which is favorably oriented with respect to the loading axis grew at the expense of the second, non-favorable oriented domain. The different slopes of Cl 1 in the loading (segment 4) and unloading (segment 5) martensite regime, however, cannot be explained at this point and also require additional work.

Finally, the reverse transformation into austenite sets in when the critical stress for its onset is reached upon further unloading (cf. Fig. 1). Over the entire reverse transformation plateau, then, the transformation into austenite not only leads to a steady accumulation of AE energy belonging to Cl 2 (cf. Figure 5b), it is also accompanied by a steady increase in the cumulated AE energy $E_\Sigma$ of Cl 1 (segment 6). Noteworthy, the cumulated AE energy $E_\Sigma$ curve of Cl 1 increases with different slopes during the forward (segment 2) and reverse transformation plateau. Rationale for this behavior can be found in the in situ optical micrographs (Fig. 1). While the forward transformation is characterized by the nucleation and growth of a dominant martensite plate, numerous nucleation sites are formed with the onset of the reverse transformation. Hence, the increased number of austenite/martensite interfaces (habit planes) causes an increased number of AE signals of Cl 1, leading to a higher slope of the corresponding curve during the reverse transformation compared to the forward one. This seems to be in agreement with results found by Bonnot et al.[12] on Cu-Zn-Al SMA tested under SE tensile conditions, where also such kind of asymmetry was observed, but it was not fully clarified. The asymmetry, however, is in contradiction to Fe-based SMAs such as Fe-Mn-Al-Ni, where already in the martensite unloading regime, i.e. well before the stress plateau of the reverse transformation is reached, numerous tiny martensite plates retransform to austenite causing large amount of transient signals as shown in ref. 7.

Based on the findings obtained within the present study, Fig. 6 schematically highlights the deformation mechanisms, which have to be considered for ⟨001⟩-oriented Heusler-type Co-Ni-Ga SMA single crystals in solution-annealed condition under SE compressive loading at 100 °C. The schematic is designed to allow for rapid access to the main findings of the present work, which were already presented and discussed above. The solution-annealed and secondary phase free material state is characterized by a SE behavior with full strain recovery. This excellent functional performance is based on the thermo-elastic stress-induced phase transformation from the B2-ordered austenitic matrix to a non-modulated tetragonal martensite with $L1_0$

structure (b, c) and vice versa (f). The excellent reversibility is governed by the easy motion of the stress-induced martensite plates (all HPV plates have nearly the same orientation with respect to the loading direction, cf. b, c, f). Under the present loading conditions, i.e. compression along ⟨001⟩, all martensite plates consist of the same two martensite domain variants, i.e. a single internally-twinned CVP. In the martensite regime, where the MPT is assumed to be fully completed, twin reorientation by twin boundary motion takes place. While upon loading detwinning by the growth of the dominant martensite domain variant at the expense of second, non-favorable oriented domain is assumed, the opposite re-twinning process, i.e. the growth of a second, minor dominant domain (d, e), occurs during unloading.

## Methods

### Material

From induction melted Co-Ni-Ga ingots with a nominal chemical composition of 49Co-21Ni-30Ga (in at.%) large single crystals with sizes up to several centimeters were grown by the Bridgman technique under helium atmosphere. The specific alloy composition has been designed for enhanced functional performance, i.e. a high degree of strain recoverability[19]. Rectangular samples with dimensions of $3 \times 3 \times 6$ mm³ were electro-discharge machined (EDM) from one of the bulk crystals such that their longer (i.e. loading) axes were parallel to the ⟨001⟩ crystal direction of the austenitic phase, while normal vectors of the lateral surfaces were parallel to ⟨100⟩ and ⟨010⟩. Following EDM, the lateral surfaces of the samples were mechanically ground to remove any residue from machining. In order to obtain a single-phase material state free of any secondary phases, which can significantly affect the deformation behavior[18,24,27], all samples were initially solution-annealed. The solution-annealing treatment was conducted at 1200 °C for 12 h in sealed quartz glass tubes under argon atmosphere, followed by breaking manually the quartz tubes at ambient conditions (air). Employing differential scanning calorimetry (DSC), the characteristic transformation temperatures were determined. Single-crystalline samples with a mass of about 15 mg were prepared and investigated using a PerkinElmer DSC 8500. Figure 7 shows a characteristic DSC plot obtained at heating and cooling rates of 20 °C/min. The solution-annealed Co-Ni-Ga is characterized by martensite finish ($M_f$), martensite start ($M_s$), austenite start ($A_s$), and austenite finish ($A_f$) temperatures of −10 °C, −6 °C, 11 °C, and 14 °C, respectively.

### In situ superelastic testing

Accompanied by in situ techniques detailed in the following sections, quasi-static uniaxial compression tests were carried out at 100 °C. As can be deduced from the DSC chart in Fig. 7, the selected test temperature ensured a fully austenitic material state prior to testing of the solution-annealed samples and allowed for direct comparison with data obtained in previous studies[18,24]. Each SE single cycle test was run in displacement control at a nominal strain rate of $1 \times 10^{-3} \mathrm{s}^{-1}$ up to a maximum strain of −5% upon loading and a minimum load of -200 N for unloading. The maximum strain level was chosen with respect to the theoretical transformation strain of the ⟨001⟩ crystal orientation under compressive loading[13]. Eventually, three SE experiments, i.e. accompanied by in situ OM (s. section 2.2.1), neutron diffraction (s. section 2.2.2) and AE (s. section 2.2.3), were conducted and for each experiment, a new (virgin) single-crystalline sample was used in order to avoid effects of loading history on the stress-induced MPT behavior.

**Optical microscopy.** In situ OM experiments were performed on a servohydraulic test frame equipped with a digital microscope and a tele-zoom objective. The test temperature of 100 °C was obtained by controlled convection furnaces, while a thermocouple directly attached to one of the lateral surfaces was used for temperature control. Strains were measured employing a high-temperature extensometer with a gauge length of 12 mm. Its ceramic rods were directly

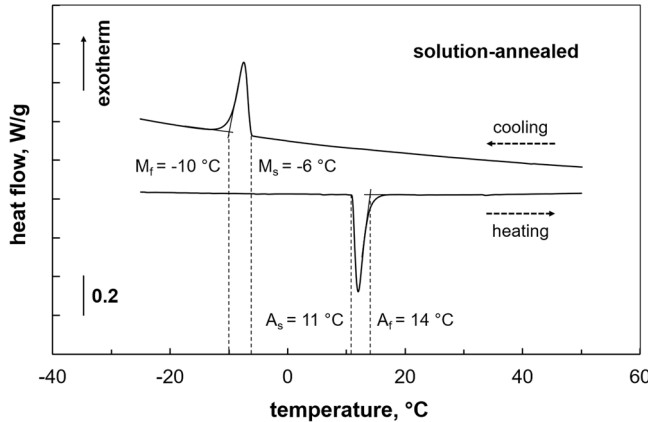

**Fig. 7 | Differential scanning calorimetry (DSC) curve of single-crystalline Co-Ni-Ga in solution-annealed condition.** The characteristic transformation temperatures upon heating and cooling, i.e. austenite start ($A_s$) and finish ($A_f$) as well as martensite start ($M_s$) and finish ($M_f$), respectively, are marked. Source data are provided as a Source Data file.

attached to the compression grips, which were treated as absolutely rigid for calculation of the nominal strain. Surface images of a representative area of up to $2.3 \times 3.0$ mm$^2$ were captured at various stages of the stress-strain curve from the center of the probed material. Following standard silicon carbide grinding down to $5\,\mu$m grit size, the lateral surface investigated by OM was additionally mechanically polished using a colloidal SiO$_2$ suspension with 0.02 µm particle size. All grinding and polishing procedures were conducted in the austenitic state of the Co-Ni-Ga SMA.

**Neutron diffraction.** In situ neutron diffraction was carried out using SXD[40], the single crystal Laue diffractometer at the ISIS spallation neutron source (Rutherford Appleton Laboratory, United Kingdom). The SE single cycle tests were performed on a miniature load frame (Kammrath & Weiss GmbH, Germany). SXD uses the TOF technique to acquire diffraction data of single-crystalline samples, employing a polychromatic neutron beam with incident wavelengths in the range of 0.2–10 Å. The beam size was set by adjusting the beam-defining jaws, ensuring that the entire sample volume was probed. In the martensite regime, diffraction data were recorded on six equatorial LiF/ZnS scintillator area detectors for 175 min during each predefined loading stage. The displacements at each loading stage were derived from the stress values obtained in the in situ OM experiments initially conducted on the servohydraulic test frame. For further details on the in situ setup of SXD, the reader is referred to ref. 23. Diffraction data were indexed and integrated using the software package SXD2001[40]. The refinement of the identified Bragg peak positions from specific lattice planes is based on optimization of the match between observed and calculated peak positions. The peak intensities are then determined based on the refined peak positions employing a single peak fitting routine as a function of time-of-flight. For more details on this procedure, the reader is referred to the *Supplementary Material* (Figs. S-1, S-2, and S-3).

**Acoustic emission.** Mechanical testing in combination with in situ measurements via AE was performed using a miniature load frame (Kammrath & Weiss, Germany). The 100 °C test temperature was obtained using a hot-air gun targeted at the center of the compression sample. The temperature was monitored via a thermocouple and an infrared thermo-camera (Vario head, Infratec, Germany). For strain measurement, digital image correlation (DIC) was applied employing a video extensometer and Veddac software (Chemnitzer Werkstoffmechanik GmbH, Germany). For this

purpose, digital images were captured using a video camera (Manta G-505, Allied Vision, Germany) and a ring illumination providing 8-bit grey scale images at a resolution of 2452 × 2056 pixels. The video camera was equipped with a tele centric optic. Images of the entire compression sample were recorded with a pixel resolution of 3.45 µm/pixel at a frame rate of 0.01 ms$^{-1}$. To ensure an adequate surface pattern contrast for DIC, the Co-Ni-Ga crystal was used and tested in grinded condition. The AE signals were recorded continuously (threshold-less) during the SE test using a piezo-electric broadband sensor (Micro F30, Physical Acoustic Corporation), which has a high sensitivity in the frequency range between 100 and 800 kHz. The AE transducer was directly attached to the gripping system of the miniature load frame. The acoustic signals were pre-amplified by 40 dB using a 2/4/6 low-noise preamplifier (Physical Acoustic Corporation, USA) and recorded with a sampling rate of 2 million data points per second via an 18-bit PCI-2 acquisition board (Physical Acoustic) with a build-in band pass filter from 30 to 1000 kHz. A comprehensive analysis of the recorded dataset was performed in the frequency domain by application of FFT with the aim to deconvolute the source function of the AE data from the transfer function including aspects of wave propagation and sensor characteristics. As a result, three different types of noise-corrected and normalized power spectral density functions $G(f)$ were obtained, which were characterized among others by two primary features: the AE energy ($E$) of the signals and their median frequency ($f_m$), which are equal to $E = \int_0^\infty G(f, t)df$ and $\int_0^{fm} G(f, t)dt = \int_{fm}^\infty G(f, t)dt$, respectively.

## Data availability
Source data are provided as a Source Data file.

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

## Acknowledgements

The authors gratefully acknowledge our partners for providing the single-crystalline material and the assistance of Keith Allum and Leoni Hübner with the experiments.

## Author contributions

C.L.: Conceptualization, Data curation, Investigation, Visualization, Writing—original draft A.W.: Conceptualization, Data curation, Methodology, Formal Analysis, Funding Acquisition, Writing—original draft R.L.: Investigation, Methodology, Formal Analysis A.R.: Investigation, Data curation, Methodology, Formal Analysis T.P.: Investigation, Visualization M.J.G.: Methodology, Validation P.K.: Funding Acquisition, Writing—review & editing W.W.S.: Funding Acquisition, Supervision H.B.: Resources, Writing—review & editing H.S.: Methodology, Validation, Funding Acquisition, Writing—original draft T.N.: Resources, Supervision, Writing—review & editing

## Funding

A.W. and P.K. disclose support for the research of this work from Deutsche Forschungsgemeinschaft (DFG) [grant number 449930948]. H.S. discloses support for publication of this work from Czech Science Foundation [grant number 24-10334S] and Czech Ministry of Education, Youth and Sports [project FerrMion, CZ.02.01.01/00/22_008/0004591, co-funded by European Union]. Open Access funding enabled and organized by Projekt DEAL.

## Competing interests

The authors declare no competing interests.
