## [Transparent Peer Review file · Nature Communications]

On the origin of acoustic emission in the stress-induced martensite regime of shape memory alloys

Corresponding Author: Dr Christian Lauhoff

Version 0:

Reviewer comments:

Reviewer #1

(Remarks to the Author)

Remarks to the Authors

The noteworthy results of this manuscript are:

- o detecting the boundary motion of twin-related stress-induced martensite plates in the elastic deformation region of martensite;

- o correlating the in situ investigations, of the structural and properties changes that accompany a compression loading-unloading cycle applied to a $\langle 001 \rangle$ -oriented single crystalline $\text{Co}_{49}\text{Ni}_{21}\text{Ga}_{30}$ specimen, by acoustic emission signals, optical microscopy observations and neutron radiation experiments, with twinning-controlled deformation in the austenitic field at 100 °C;

- o contributing to a deeper understanding of deformation and degradation mechanisms that occur in other SMA systems. The paper reports and discusses some phenomena that accompany the superelastic deformation of a $\langle 001 \rangle$ -oriented single-crystalline $\text{Co}_{49}\text{Ni}_{21}\text{Ga}_{30}$ specimen, during a compression loading-unloading cycle. Due to the in situ character of optical microscopy analysis, neutron radiation observations, acoustic emission records and their correlation with isothermal compression loading-unloading in austenitic phase, as well as owing to the throughout discussion of the investigated phenomena, the present study could gain marked significance for the Shape Memory community.

The DSC chart revealed a reversible martensitic transformation, which is the prerequisite of the shape memory effect and hereby thermal memory occurrence. The compression loading-unloading test was performed on a $\langle 001 \rangle$ -oriented single crystalline $\text{Co}_{49}\text{Ni}_{21}\text{Ga}_{30}$ specimen in austenitic state (at a temperature higher than A_s , determined from DSC thermogram) and revealed a perfect superelastic behavior. In situ optical microscopy analysis, neutron diffraction measurements and acoustic emission records were synchronized with mechanical testing. The discussion is based on speculating the contrast similarity between the two differently colored regions from optical microscopy micrographs. Assuming that the two regions correspond to two twin-related stress-induced martensite plate variants, the entire discussion is valid.

In the opinion of the present reviewer, a surface profile image obtained by atomic force microscopy (AFM) would definitely prove the validity of the interpretation of optical microscopy micrographs.

In order to support the entire discussion, AFM micrographs would be welcome, accompanied by surface profile images, proving that the two illustrated martensite plate variants, V1 and V2, are twin-related and that V2 is replaced by V1 during loading and that V1 is detwinned during unloading being replaced by V2.

The methodology is sound and, if sustained by AMF, it would meet all the expected standards in the field, providing enough details enabling the experiments to be reproducible. Figure 6 has to be revised to make dotted circles visible.

Reviewer #2

(Remarks to the Author)

The manuscript present results of experiment focusing stress induced martensitic transformation in $[001]$ oriented single crystal of CoNiGa shape memory alloy deformed in compression at 100 °C. The surface of the sample is observed by optical microscopy, volume fraction of martensite twins was evaluated by Laue diffraction of neutrons and acoustic emission generated by the activated deformation processes was in-situ evaluated during the compression test

In the opinion of the referee, key result is that the authors were able to associate the experimentally observed low median frequency acoustic events with twinning in martensite during the unloading branch of the compression test (it shall be mentioned in Conclusions). If true, this may have an important impact on AE experimentation on SMAs in the field.

However, the presentation and interpretation of the results in the manuscripts raises lots of questions and issues that needs to be resolved before the manuscript can be recommended for publication.

Questions concerning in-situ mechanical experiment

1. In situ experiment means that the AE and OM is performed during the experiment at ISIS, which does not seem to be the case. It needs to be clearly described in the manuscript how the in-situ experimentation is meant.

2. There is a problem with the OM. While neutron diffraction and AE takes average information, the tested sample has face 3*6mm but figure 2 shows only 1x1.5mm area. Direct comparison is thus questionable. Would it be possible to show larger area?

3. There is a stress plateau from strain 1% up to 2.5% and than the stress starts to increase. The referee has a suspicion that the forward MPT changes its character, probably a single habit plane mode changes into multi habit plane mode or even different variant is induced at the corner. Is that possible? If yes, it would question ND and AE results.

Questions concerning neutron diffraction experiment

1. The results of neutron diffraction experiment are limited to 4 datapoint in figure 4b. Given the importance of this result for the proposed interpretation, better presentation is necessary

2. Twinning in SMA single crystals under mechanical loads was already studied by neutron diffraction – these works need to be cited in chapter 3.2. Otherwise, it looks like that the authors perform first ever ND experiments on twinning. It will show up that earlier ND experiments already observed twinning on elastic unloading of martensite crystals

Questions concerning the AE experiment

1. Why the median frequency starts to increase only from the middle of the stress plateau (from time 40s and not from 20s)? why this rise in median frequency (35s) is preceded by the peak of AE energy (50s)

2. Why AE energy is lower during the reverse MPT than during the forward MT. What the AE peaks during the forward and reverse MPT correspond to?

3. The cluster 2 actually consists of tree clusters why they were not distinguished?

4. Figure 5d needs to be plot also with stress and strain on the x-axis

Questions concerning the low frequency AE emission

1. If the of the low frequency cluster of AE is due to twinning in stress induced martensite, the rate of increase of the cumulative energy of the low frequency cluster with time (red curve in fig 5d, further only rate) shall increase upon tensile loading beyond the end of the plateau (No constraint from habit plane), but it decreases. Why?

2. The rate is high upon elastic unloading in martensite and decreases during the reverse MPT – this can be rationalize by decrease of the volume fraction of martensite. But the red curve needs to be plot with respect to stress and strain to be able to compare that. Please draw such figure

3. The rate however remains the same upon elastic unloading in austenite Why?

4. Why there is such pronounced difference in the rate between the forward and reverse martensitic transformation?

5. If the of the high frequency cluster of AE is due to martensitic transformation (blue curve in fig 5d, further only rate), why it is lower during the reverse MT?, why it has steps? and why total amount of energy released is lower? The blue curve needs to be plot with respect to stress and strain to be able to compare that. Please draw such figure.

Questions concerning the deformation mechanism

1. The experiment suggests that twinning in martensite occurs during reverse MPT on unloading. That means that habit plane orientation changes with decreasing stress. Is that the case?

2. It is written on page 12 that “However, all habit planes formed during forward and reverse, transformation are parallel to each other, i.e. they feature the same orientation with respect to the loading direction, which is horizontal in Fig. 2.” But

evidently this is the case only in first approximation. In fact the traces of habit planes in fig. 2 are frequently not single lines. These small changes of habit plane trace shall be evaluated.

3. Why there is such pronounced difference in twinning activity between the forward and reverse martensitic transformation?

4. Why no attention is given to the potential change of habit plane with varying stress if PTMC theory relates the habit plane interface to the volume fraction of twins in martensite and experimental results suggest activity of twinning during the martensitic transformation. There are theoretical works on this topic in the literature.

Questions concerning interpretation and discussion

1. Since no plastic strains is generated during the loading-unloading cycle, there is only marginal dislocation slip and it is unlikely the AE signal can be ascribed to it. The referee disagrees with the text on lines 420-425. Please provide a better explanation than dislocation slip in austenite. Dislocation slip in austenite in CoNiGa can be safely excluded at these stresses, particularly on unloading. If you do not know the source of this signal, just do not explain it.

2. Figure 3 needs to show also habit plane trace

3. The text on page 22 "After the forward transformation plateau, once CI 2 remains more or less constant, CI 1 shows a significant increase during both further loading in the martensite regime up to the maximum strain level of -5% (segment 4)" is incorrect - the corresponding rate in Fig. 5c does not increase in segment 4 more than before.

4. Since the load and face orientations of the crystal deformed in compression and volume fraction of twins (PTMC and experiment) are known, it is possible to plot exact traces of habit planes and twin planes on one of the sample faces (or even on a prism representing the sample). Such information needs to be included into figure 6

5. Page 25 lines 500-503 "the authors claim that "The excellent reversibility is governed by the easy motion of the stress-induced martensite plates of a single HPV system (all martensite plates have the same orientation with respect to the loading direction, cf. b, c, f), since only fraction of surface on a single face was observed by OM it is possible that another variant appears elsewhere. The increasing stress in figure 2 beyond 2.5% strains suggest that possibility.

6. Page 25 lines 503-506 "However, even though there is a lack of available slip systems in the $\langle 001 \rangle$ crystal direction under uniaxial loading, irreversible processes cannot be fully excluded, which in turn do not compromise the reversibility. Minor dislocation activities seem to appear in the B2 ordered parent phase over the entire SE cycle (a, b, f, g)." Dislocation slip creates slip traces and unrecovered plastic strain. The authors did not provide any evidence for dislocation slip in the experiment. Therefore, they cannot argue with it to interpret the experimental results

7. Page 25 lines 510-513 "In the martensite regime, where the MT is assumed to be fully completed, twin re-orientation by twin boundary motion takes place. While upon loading detwinning by the growth of the dominant martensite domain variant at the expense of second, non-favorable oriented domain is assumed,..." Although the activation of detwinning on forward loading is logical, no experimental evidence was provided for that.

Comments to the Conclusions

1. Page 26 lines 522-524 "The complementary findings obtained from the imaging, diffraction and acoustic measurements allowed a direct correlation between AE signals and the deformation behavior, i.e. dislocation activities, stress-induced MPT and twinning processes." This claim is not fully supported by the results (dislocations, only fraction of surface observed)

2. Page 26 lines 525-536 The text discusses low AE energy signals but it is probably more important to mention that the authors associate the low median frequency signal with detwinning process. The referee understands that low AE energy signal is low median frequency signal, yet still this needs to be mentioned in Conclusions

Reviewer #3

(Remarks to the Author)

This particular work contains interesting results regarding acoustic emission caused in general by stress-induced martensitic transformation. It is an original one and shows the significance to the field, especially while it is related to methodology of the acoustic signals evaluation. Authors considered how microstructural features are evolving in the martensite state, which is important for shape memory alloys in general and for superelastic behavior in particular. In the text I left a number of comments and corrections that might be of help (all of it is in attached file - comments, inserts etc in pdf). Some required ones are related to some works published previously. Yet, these papers that are now missing in the text of the present paper are important but in general their absence do not prohibit publication. On the other hand, I'd like to insist on a proper corresponding revision. Also it can be good for the paper to include actual neutron diffraction patterns with proper indexing and showing exactly what are those variants that change their volume fraction upon unloading (that is also related to

reproducibility issue). Some additional optical microscopy pictures somewhere between point e and f in Fig. 2 might also make obtained results more clear. My conclusion is that this paper will be ready for publication after revision.

Version 1:

Reviewer comments:

Reviewer #1

(Remarks to the Author)

You responded to most of my comments in a satisfactory way. From my point of view, the paper can be published now.

Reviewer #2

(Remarks to the Author)

The authors responded to all questions of all 3 referees and made appropriate changes in the manuscript which resulted in improvement.

However, I complained in my previous comments that "interpretation of the results in the manuscripts raises lots of questions and issues that need to be resolved before the manuscript can be recommended for publication" and this unfortunately did not really happen. Many issues still remain in the revised manuscript. Let me raise only two, which I consider to be most important

As concerns my questions concerning AE experimental results (4-14), the authors were not able to provide answers to most of them. Although that is understandable (results of AE experiments are very difficult to interpret), I am not sure if this is appropriate for a nature paper

More problematic is the response of the authors to the questions 15-18 regarding the experimental evidence for twinning obtained by AE and ND. The referee is satisfied neither with the response in the rebuttal nor with the changes made in the manuscript. If tetragonal single crystal is loaded in compression, the martensite phase appearing behind the propagating habit plane is composed of martensite variants V1 and V2 providing same strain along the load axis (see Fig. 3). Because of that, there is no driving force for activation of detwinning on loading and twinning on unloading in the martensite state. If the authors have followed the referee suggestion in comment 20 "Figure 3 needs to show also habit plane trace", they would figure out this.

In spite of that, ND experiment detects small change of the peak intensities on unloading, which provides an evidence that detwinning occurred on unloading between the two measurement points. This is probably due to slight misalignment in the experiment, which is common in single crystal experiments. If the authors make similar experiment in tension, two martensite variants would provide very different strain to the load axis, there will be driving force for detwinning on loading and unloading in martensite state. They should have realized this simple consequence of deformation geometry and the experiment should have been performed in tension.

Second issue concerns the author's response to the comment related to the change of the habit plane trace orientation observed by OM upon loading and unloading. The authors made changes in the manuscript which acknowledge the experimentally observed changes of the habit plane trace. That is fine. But it seems they did not figure out the reason for it (or at least they did not explain it properly in the manuscript)

The authors deal with this on page 13 line 273-281. „The latter, in turn, is seen to be based on incompatibilities and further accommodation processes between the austenitic parent phase and the twinned martensite to maintain a strain compatible habit plane interface. Theoretical work on this topic is available in literature, for example in Ref. [28], where the classical phenomenological theory of martensite crystallography (PTMC) [29] was extended to include the effect of elastic deformation of the austenite and martensite lattices due to external stress. However, for the discussions on the martensite variant selection, i.e. the number of HPV systems, these slight orientation differences between the habit plane traces can be neglected.“

However, the above text is rather confusing, as it mixes different phenomena and does not explain the experimental observation

Considering only MT (i.e. no dislocation slip), according to the PTMC theory, habit plane orientation is linked to the shape strain of the martensite. If shape strain of the martensite changes slightly by detwinning on loading and significantly by twinning on unloading, habit plane interface propagating during the reverse MT on unloading must differ (lie on a different austenite plane) from that which propagated on forward loading. Or there would have to be other mechanism assuring strain

compatibility at the propagating habit plane interface interface (dislocation slip or additional twinning prior the habit plane in martensite) propagating during the reverse MT. If authors believe that PTMC theory is correct, they have to acknowledge this effect. Of course, the effect is small because the amount of twinning is small due to the absence of the driving force for it. However, even if the effect is small, it is a consequence of a nature law (this referee believes that PTMC theory is right in the treatment of the requirement of strain compatibility at the habit plane interface as a nature law) and nature laws shall not be neglected in nature papers.

The authors mention the effect of stress on habit plane orientation of stress induced MT elaborated in Ref. [28]. Yes, that is quite possible too, but this is a very different effect, which this referee did not have in mind while commenting on this. Classical PTMC theory neglecting stress effects on habit plane is sufficient to explain the observed habit plane changes.

Although the two comments raised above are rather negative, the referee is not negative to the whole manuscript. Appropriate modifications in the revised manuscript may solve these issues.

Reviewer #3

(Remarks to the Author)

Considering the fact that authors have taken into account most of my remarks and did their own discussion work in the vicinity of these remarks, I suggest that with all the present changes this paper can be published.

Version 2:

Reviewer comments:

Reviewer #2

(Remarks to the Author)

The authors solved the first issue I raised in previous comments with the added supplement, though if I understand correctly, there is no experimental evidence for those fine martensite plates predicted by the theoretical calculations.

Nevertheless, if the theoretical prediction describes reality, those fine martensite plates growing during the forward loading would generate major acoustic events of largest amplitude and energy, in my experience.

Concerning the second issue, the theoretical calculation in the supplement still disregards the effect of elastic strains in austenite and martensite on strain compatibility at the habit plane orientation. But this is fine, since the effect of grip constraint is probably even more pronounced.

I am happy to recommend the revised manuscript for publication.
Below are comments to few bugs I spotted while reading the revised manuscript.

Page 12/line 239 near (101) plane of the austenite

Page 14/line 277 all habit planes that formed during forward and reverse transformation can be considered

Page 17/line 347 there would be small variations of martensite lattice rotations,

RESPONSE TO REVIEWERS' COMMENTS

We thank the reviewers for their positive feedback. The changes we made to address their valuable comments and useful suggestions are explained in our response to the reviewers and are marked **yellow** in the "marked up manuscript" version, respectively.

Reviewer #1: The noteworthy results of this manuscript are:

- detecting the boundary motion of twin-related stress-induced martensite plates in the elastic deformation region of martensite;
- correlating the in situ investigations, of the structural and properties changes that accompany a compression loading-unloading cycle applied to a $\langle 001 \rangle$ -oriented single crystalline $\text{Co}_{49}\text{Ni}_{21}\text{Ga}_{30}$ specimen, by acoustic emission signals, optical microscopy observations and neutron radiation experiments, with twinning-controlled deformation in the austenitic field at 100 °C;
- contributing to a deeper understanding of deformation and degradation mechanisms that occur in other SMA systems.

The paper reports and discusses some phenomena that accompany the superelastic deformation of a $\langle 001 \rangle$ -oriented single-crystalline $\text{Co}_{49}\text{Ni}_{21}\text{Ga}_{30}$ specimen, during a compression loading-unloading cycle. Due to the in situ character of optical microscopy analysis, neutron radiation observations, acoustic emission records and their correlation with isothermal compression loading-unloading in austenitic phase, as well as owing to the throughout discussion of the investigated phenomena, the present study could gain marked significance for the Shape Memory community.

The DSC chart revealed a reversible martensitic transformation, which is the prerequisite of the shape memory effect and hereby thermal memory occurrence. The compression loading-unloading test was performed on a $\langle 001 \rangle$ -oriented single crystalline $\text{Co}_{49}\text{Ni}_{21}\text{Ga}_{30}$ specimen in austenitic state (at a temperature higher than A_s , determined from DSC thermogram) and revealed a perfect superelastic behavior. In situ optical microscopy analysis, neutron diffraction measurements and acoustic emission records were synchronized with mechanical testing. The discussion is based on speculating the contrast similarity between the two differently colored regions from optical microscopy micrographs. Assuming that the two regions correspond to two twin-related stress-induced martensite plate variants, the entire discussion is valid.

In the opinion of the present reviewer, a surface profile image obtained by atomic force microscopy (AFM) would definitely prove the validity of the interpretation of optical microscopy micrographs.

In order to support the entire discussion, AFM micrographs would be welcome, accompanied by surface profile images, proving that the two illustrated martensite plate variants, V1 and V2, are twin-related and that V2 is replaced by V1 during loading and that V1 is detwinned during unloading being replaced by V2.

The methodology is sound and, if sustained by AFM, it would meet all the expected standards in the field, providing enough details enabling the experiments to be reproducible. Figure 6 has to be revised to make dotted circles visible.

RESPONSE #1: The authors appreciate the positive feedback of the reviewer and thank her/his for the valuable comments! The pure suggestions for alternative expressions that were raised by the reviewer in the attachment have been taken over in the revised version of the manuscript. All other remarks which need (more) clarification are redlined in the following:

1. Line no. 247-250: This elastic unloading regime ... was chosen ... since in this region both no superimposed effects by a simultaneous reverse transformation of martensite into austenite were to be expected (cf. Fig. 2) and, at the same time, the intensity of AE signals was found be unexpectedly high ...

RESPONSE #1.1: The authors thank the reviewer for her/his alternative wording raised in the PDF file attached. Unfortunately, the suggestion does not reflect the authors' intended message. In order to improve language and, at the same time, to avoid any misleading interpretation, the original sentence has been revised to: "This unloading regime of the martensite was selected for the neutron diffraction experiments due to the following two reasons: (1) Superimposed effects by a simultaneous reverse transformation of martensite into austenite were not to be expected and, thus, did not affect the evaluation of a twin re-orientation mechanism. (2) In comparison to the loading regime of martensite (s. segment 4 in Fig. 6b), the intensity of AE signals was found to be unexpectedly higher (s. segment 5 in Fig. 6b), suggesting more pronounced twin re-orientation activities in this region (s. discussions in the remainder of the text)." (Page 15)

2. Line no. 295-296: The present reviewer suggests that, if possible, AFM micrographs with corresponding surface profiles should be recorded in points (1) and (2).

RESPONSE #1.2: The authors thank the reviewer for this valuable comment and can fully understand her/his suggestion. Indeed, AFM is a valuable technique to investigate the martensitic microstructure by surface profiles. Unfortunately, an adequate setup allowing for an *in situ* SE compression experiment at 100 °C accompanied by AFM is not available and, thus, the additional results inquired by the reviewer cannot be provided.

Nonetheless, the present authors would like to point to an earlier study from 2008 by Niklasch et al. [1]. In that study, the authors investigated the magnetic domain structure in $\langle 001 \rangle$ -oriented single-crystalline CoNiGa by (high-resolution) MFM. Under compressive loading, Niklasch et al. [1] found martensite plates being internally twinned, which is excellent in line with the present findings. However, the corresponding twins, i.e. the individual martensitic domain structures forming a CVP system, are less than 1 μm in thickness, i.e. in a range that cannot be resolved by the OM setup employed. While AFM/MFM is not available as mentioned before, the neutron diffraction results presented in the manuscript clearly allow for a reliable and quantitative assessment of the twinning state. For more clarity and visualization of twinned martensite in $\langle 001 \rangle$ -oriented single-crystalline CoNiGa under compression (even though not directly provided in the present study), the authors have carefully revised the text and, now, refer the reader to the study by Niklasch et al. [1]. (Page 12)

Reviewer #2: The manuscript present results of experiment focusing stress induced martensitic transformation in [001] oriented single crystal of CoNiGa shape memory alloy deformed in compression at 100 C. The surface of the sample is observed by optical microscopy, volume fraction of martensite twins was evaluated by Laue diffraction of neutrons and acoustic emission generated by the activated deformation processes was in-situ evaluated during the compression test.

In the opinion of the referee, key result is that the authors were able to associate the experimentally observed low median frequency acoustic events with twinning in martensite during the unloading branch of the compression test (it shall be mentioned in Conclusions). If true, this may have an important impact on AE experimentation on SMAs in the field.

However, the presentation and interpretation of the results in the manuscripts raises lots of questions and issues that needs to be resolved before the manuscript can be recommended for publication.

Questions concerning in-situ mechanical experiment:

1. In situ experiment means that the AE and OM is performed during the experiment at ISIS, which does not seem to be the case. It needs to be clearly described in the manuscript how the in-situ experimentation is meant.

RESPONSE #2.1: The authors thank the reviewer for this valuable and helpful comment. It is of utmost importance that the reader is able to relate to the *in situ* experimentation. The reviewer is fully right that AE and OM were not performed during the experiment at ISIS. The setup at ISIS does not allow for such an experiment. In the current study, the superelastic testing under compression was accompanied by a single technique at a time, i.e. either OM, or AE or neutron diffraction. For this experimentation, a total of three virgin Co-Ni-Ga single-crystalline samples were used, i.e. one for each *in situ* experiment. Even though all three techniques were not employed at the same time, in the authors' opinion, the wording "*in situ* testing" is fully legitimate at this point. In light of the reviewer's query, the section 2.2 has been carefully revised and further details about the experimental procedure, especially the number of tested samples, have been added for more clarity. (Pages 7 and 8)

2. There is a problem with the OM. While neutron diffraction and AE takes average information, the tested sample has face 3*6mm but figure 2 shows only 1x1.5mm area. Direct comparison is thus questionable. Would it be possible to show larger area?

RESPONSE #2.2: The reviewer is fully right that the optical micrographs presented do not capture the entire sample. However, Fig. 2 does not only assess quite localized areas of 1 mm x 1.5 mm. In fact, it shows representative areas of 2.3 mm x 3.0 mm from the midsection of the sample. The dimensions of the compression sample were 3 mm x 3 mm x 6 mm and, thus, the area probed was more than one third of the entire sample.

Beyond that, the authors are fully aware that the martensitic microstructure can be affected by applied constraints owing to the compression grips. However, the neutron diffraction results are clear evidence that such artefacts can be neglected in the present study. In contrast to the optical micrographs, the neutron beam covered the entire sample, i.e. volumes that might be affected by the grips as well. Nonetheless, no evidence of other CVP systems as described in the manuscript were found. This means that either the grips have not influenced the transformation behavior or if additional martensitic domain variants have been formed in the vicinity of the grips (not observed by OM analysis), then their volume fractions were of such marginal magnitude that they were rendered indistinguishable from the background signal.

Eventually, the findings obtained by OM and neutron diffraction are fully consistent with each other. While the optical micrographs reveal only martensite plates with one single global orientation (Fig. 2), i.e. one single HPV system, the neutron diffraction results show the presence of a single CVP system, of which the martensite plates (the HPV system) consist. Thus, direct comparison is fully justified as it was already done in the previous studies [2–4]. In light of the reviewer's criticism, however, details about the area and volume sizes probed by OM and neutron diffraction, respectively, have been added to the manuscript. (Pages 8 and 9)

3. There is a stress plateau from strain 1% up to 2.5% and then the stress starts to increase. The referee has a suspicion that the forward MPT changes its character, probably a single habit plane mode changes into multi habit plane mode or even different variant is induced at the corner. Is that possible? If yes, it would question ND and AE results.

RESPONSE #2.3: This is an interesting point mentioned by the reviewer. Indeed, there is a slight increase in stress from 2.5% compressive strain onwards. A similar behavior has been also documented in one of the previous studies by some of the present authors [2], however, the underlying microstructural mechanism can/could not be resolved so far. In both studies, i.e. the former one [2] and the present work, *in situ* OM analysis just reveals a single HPV configuration for the solution-annealed (001)-oriented single-crystalline Co-Ni-Ga samples under compressive loading. As already stated in the previous response #2.2, the optical micrographs cover a representative area from the midsection of the compression samples, while information from other surfaces or the edges are not available. Hence, a multi HPV mode with habit planes of different orientations, potentially affecting the stress plateau, cannot be completely ruled out by the OM analysis at this point. However, the authors would like to emphasize that the bulk (volume) information from neutron diffraction, even though obtained from another solution-annealed sample, perfectly match the OM findings. In fact, a single CVP has been detected by diffraction, indicating a single HPV configuration as shown by OM. Besides this, the main objective of the present study was to shed light on the origin of AE signals in the unloading martensite regime. This origin, i.e. re-twinning by martensite domain variant re-orientation, can only be deduced from the neutron diffraction results, providing bulk information from the entire sample volume. The OM images, in turn, are just for visualization of the mesoscopic stress-induced MPT behavior and thus more clarity for the readership.

Questions concerning neutron diffraction experiment

4. The results of neutron diffraction experiment are limited to 4 datapoint in figure 4b. Given the importance of this result for the proposed interpretation, better presentation is necessary.

RESPONSE #2.4: The authors agree with both this reviewer and reviewer #3 (s. comment #3.6) that the presentation of the neutron diffraction results has been kept to an absolute minimum in the manuscript. At this point, the authors would like to mention that the fundamental mechanism, i.e. twinning upon unloading in the martensite regime, is absolutely clear from the few datapoints shown in Fig. 4b. Nonetheless, the authors can understand the criticism raised by the reviewers. For more traceability and easier reproducibility of the findings presented, hence, further details about the analysis routine of the neutron diffraction data including diffraction patterns and proper indexation following refinement are presented and described in the Supplementary Material now.

5. Twinning in SMA single crystals under mechanical loads was already studied by neutron diffraction – these works need to be cited in chapter 3.2. Otherwise, it looks like that the authors perform first ever ND experiments on twinning. It will show up that earlier ND experiments already observed twinning on elastic unloading of martensite crystals

RESPONSE #2.5: The authors appreciate this valuable comment. It was not the intention of the authors to imply that the present study is the first assessing twinning by neutron diffraction experiments. In light of the reviewer's comment, the authors have revised the manuscript and now clearly point to existing literature in this field. (Page 15)

Questions concerning the AE experiment

6. Why the median frequency starts to increase only from the middle of the stress plateau (from time 40s and not from 20s)? why this rise in median frequency (35s) is preceded by the peak of AE energy (50s)

RESPONSE #2.6: The authors thank the reviewer for this valuable remark. The median frequency starts to increase right from the beginning of the test (s. first region (i) in Fig. 5b) and

continuously raise in the following region (ii) of Fig. 5b, where however a steeper increase is observed at around 40s. This is caused by the fact that with progressing increase in stress and strain the ongoing microstructural processes cause a significant reduction in the mean free path/volume for either the movement of dislocations (if any), the formation of new martensitic regions and the movement of the austenite/martensite interfaces (habit planes). The text related to Fig. 5 was revised accordingly. (Pages 20 and 21)

7. Why AE energy is lower during the reverse MPT than during the forward MT. What the AE peaks during the forward and reverse MPT correspond to?

RESPONSE #2.7: The authors thank the reviewer for this valuable comment. On the one hand, there are transient (burst-type) signals with high AE amplitudes and high AE energies, which are attributed to cluster 2 (blue). These signals result from the MPT and, thus, have been recorded solely in the regions of the forward and reverse transformation (stress plateaus, cf. sections 2 and 6 in Fig. 6b of the revised manuscript, respectively). On the other hand and overlapping with these transient signals, so-called continuous signals occur with lower AE amplitudes and lower AE energies (cluster 1, red), which are related to microstructural processes operating at lower velocities such as dislocation movement (if any), movement of austenite/martensite interphase boundaries (habit planes) and movement of twin boundaries (twin re-orientation).

Noteworthy, the evolution of CI 1 and CI 2 shows significant differences. For the latter, a staircase like appearance of the cumulated AE energy (blue curve in Fig. 6b of the revised manuscript) can be seen. This is related to the individual AE energy value of each cluster element. It is nicely visible that each larger step in the blue curve is related to a high energy transient signal in the AE data stream (Fig. 6b in the revised manuscript). In other words, while the cumulated number of cluster elements of CI 2 is quite low (Fig. 6a in the revised manuscript), the individual AE energy value of each cluster element is high, resulting in a staircase shaped evolution of the cumulated AE energy of CI 2. In contrast, the red curve of CI 1 does not show these steps since here the related AE energy values per individual cluster element are very small, leading to a smooth progression.

Beside this difference, moreover, there are pronounced differences in the evolution of the clusters, depending on the loading path (direction). On the one hand, CI 2 features different total amounts of released cumulated AE energy between the forward and reverse MPT. On the other hand, different slopes in the cumulated AE energy of CI 1 can be seen in the stress-induced martensite regime, i.e. beyond the forward transformation plateau. While the asymmetry between the loading and unloading path in the martensite regime (cf. red curve of CI 1 in sections 4 and 5 of Fig. 6b, respectively) remains unresolved at this point (s. response #2.17), the difference in CI 2, i.e. the different amounts of released cumulated AE energy between the forward and reverse MPT (cf. blue curve in sections 2 and 6 of Fig. 6b, respectively) can be directly deduced from the macroscopic optical observations in Fig. 2 of the revised manuscript. As can be seen, during the forward transformation the austenitic parent phase transforms into a dominant martensite plate, whereas numerous tiny austenite plates form from the stress-induced martensite during the reverse transformation upon unloading. At this point, it is important to note that the released AE energy of a transformation event depends on the dimensions of the martensite/austenite plates and the involved volume fraction, i.e. as larger the volume fraction as higher the energy release. Since the austenitic regions upon unloading are smaller in their dimensions (volume), the cumulated AE energy of the signals related to the reverse transformation is less compared to the forward transformation even though the total transforming volume during forward and reverse MPT is the same (s. fully reversible stress-strain response). This asymmetry was already described for $\langle 001 \rangle$ -oriented Fe-Mn-Al-Ni SMA in [5].

In light of the reviewer's query and for more clarity on the AE results, the manuscript has been carefully revised by adding more details about the two clusters 1 and 2 and their specific features mentioned above. Furthermore, a new figure, namely Fig. 6, has been added, showing not only the cumulated AE energy (Fig. 6b, former Fig. 5d) but also the cumulated number of cluster elements over the entire SE cycle (new Fig. 6a). (Pages 22 – 24)

8. The cluster 2 actually consists of three clusters why they were not distinguished?

RESPONSE #2.8: This is an interesting query. At this point, however, the authors do not have a satisfying answer to this. The underlying Kullback-Leibler divergence criterion of the applied adaptive sequential k-means algorithm was not able to separate cluster 2 into three individual subclusters. Even if it would be possible, there would be no physical explanation for three individual clusters belonging to fast processes. The authors tried to separate the cluster elements according to loading and unloading path, but also without success. One possible explanation could be that the formation of the CVP system within the martensite plates of the HPV system – so-called twinned martensite – will cause burst-type signals. However, since the formation of HPV and CVP occurs simultaneously, they cannot be separated from each other, at least at the moment. A comment accordingly has been added to the text. (Pages 21 and 22)

9. Figure 5d needs to be plotted also with stress and strain on the x-axis

RESPONSE #2.9: In light of the reviewer's criticism, the authors added an additional x-axis for strain (top) in Figs. 5a and b and also in the new Figs. 6a and b. (Pages 19 and 22, respectively)

Questions concerning the low frequency AE emission

10. If the rate of the low frequency cluster of AE is due to twinning in stress induced martensite, the rate of increase of the cumulative energy of the low frequency cluster with time (red curve in fig 5d, further only rate) shall increase upon tensile loading beyond the end of the plateau (No constraint from habit plane), but it decreases. Why?

RESPONSE #2.10: The authors thank the reviewer for this valuable comment. Obviously, it was not clearly enough described in the text. The low energy/low median frequency cluster 1 (red) is formed by three different mechanisms operating all at lower velocities: (1) dislocation movement (if any), (2) movement of austenite/martensite interphase boundaries (habit planes) and (3) movement of twin boundaries (twin re-orientation). Nearby the end but still within the stress plateau, then, the slope of the cumulated AE energy of CI 1 decreases (s. segment 3 in Fig. 6b). This can be rationalized by the fully completed stress-induced MPT (from a macroscopically point of view, cf. Fig. 2d). As a result, habit plane motions are supposed to be fully diminished and, thus, no microstructural events are contributing to AE signals of CI 1 in this stage. In light of the reviewer's remark, the text has been carefully revised for more clarity. (Pages 25 and 27)

11. The rate is high upon elastic unloading in martensite and decreases during the reverse MPT – this can be rationalized by decrease of the volume fraction of martensite. But the red curve needs to be plotted with respect to stress and strain to be able to compare that. Please draw such figure

RESPONSE #2.11: The authors would like to refer to comment #2.9. Additional strain axis was added to Figs. 5a and b as well as Figs. 6a and b. (Pages 19 and 22, respectively)

12. The rate however remains the same upon elastic unloading in austenite Why?

RESPONSE #2.12: This is indeed a good question. However, at the moment the authors do not have a satisfying answer to this. Since interface movement can be safely ruled out in this regime, movement of dislocations were initially suggested to contribute to CI 1 (red curve in Fig. 6b of the revised manuscript) during unloading of austenite. Following the concern of this reviewer about slip under the present loading conditions (cf. response #2.19), however, dislocation activities in general are no longer considered in this study and a detailed analysis of the AE signals in the elastic austenite regime has to be subject of future work.

13. Why there is such pronounced difference in the rate between the forward and reverse martensitic transformation?

RESPONSE #2.13: The authors thank the reviewer for this valuable question, which has been already answered in response #2.7. Indeed, there is a distinct asymmetry in the AE signals for the forward and reverse MPT (cf. plateau regions 2 and 6 in Fig. 6b of the revised manuscript, respectively). Such asymmetry in the MPT behavior between the loading and unloading path is already known from other SMAs where it is even more pronounced [5]. For details on the underlying mechanism and the revisions made to address the reviewer's query, please see response #2.7.

14. If the of the high frequency cluster of AE is due to martensitic transformation (blue curve in fig 5d, further only rate), why it is lower during the reverse MT?, why it has steps? and why total amount of energy released is lower? The blue curve needs to be plot with respect to stress and strain to be able to compare that. Please draw such figure.

RESPONSE #2.14: Again, the answers on this valuable comment are already given in response #2.7 and #2.9, respectively, where further explanations on the evolution of CI 2 (blue curve in Fig. 6b of the revised manuscript) and the details on the revisions to address the reviewer's questions are illustrated.

Questions concerning the deformation mechanism

15. The experiment suggests that twinning in martensite occurs during reverse MPT on unloading. That means that habit plane orientation changes with decreasing stress. Is that the case?

RESPONSE #2.15: The authors thank the reviewer for this question, however, this is not the case. As already stated in the original draft at several points (e.g. on pages 5 and 19), twinning refers to "twin boundary motion causing the growth of one martensite domain variant at the expense of the other one". Changes in habit plane orientation would mean that a second habit plane variant (HPV) system comprising a second CVP system arise. However, the later can be clearly ruled out from the OM and neutron diffraction results, showing a single HPV system and a single CVP.

16. It is written on page 12 that "However, all habit planes formed during forward and reverse, transformation are parallel to each other, i.e. they feature the same orientation with respect to the loading direction, which is horizontal in Fig. 2." But evidently this is the case only in first approximation. If fact the traces of habit planes in fig. 2 are frequently not single lines. These small changes of habit plane trace shall be evaluated.

RESPONSE #2.16: The authors appreciate this valuable comment by the reviewer and fully agree with her/his opinion. In first approximation, the traces of the habit planes are single lines. On closer inspection, however, a slight splitting of some habit plane traces can be seen in the optical micrographs (cf. for example habit planes traces in the upper left corner of Figs. 2c, d, and g). This phenomenon is a shadow effect caused by the characteristic surface relief of a material undergoing a MPT. In light of the reviewer's valuable hint and to avoid an inaccurate wording, the manuscript has been carefully revised, pointing towards this slight splitting of the habit plane traces. (Page 14)

17. Why there is such pronounced difference in twinning activity between the forward and reverse martensitic transformation?

RESPONSE #2.17: This is an excellent point mentioned by the reviewer. Indeed, there is significant difference in the rate/slope of CI 1 (red curve in Fig. 6b of the revised manuscript) between loading and unloading in the stress-induced martensite regime. While the design of experiment only comprised neutron diffraction on the unloading path (s. also details in

response #2.25), the differences in twinning activity in relation to the direction of loading remain unresolved. This problem was clearly beyond of the scope of the present study and has to be investigated in future work.

18. Why no attention is given to the potential change of habit plane with varying stress if PTMC theory relates the habit plane interface to the volume fraction of twins in martensite and experimental results suggest activity of twinning during the martensitic transformation. There are theoretical works on this topic in the literature.

RESPONSE #2.18: The authors thank the reviewer for this valuable comment. Indeed, the PTMC theory is a valuable model to predict geometrical features of the MPT, such as habit plane characteristics or orientation relationships. Nowadays, modifications of the PTMC theory even allow for its application to stress induced martensitic transformations by accounting for the effect of elastic strains resulting from external stress fields for example, as recently shown in [6]. However, the application of the PTMC theory is out of the scope of the present study. The objective is to demonstrate the origin of the AE signals upon loading and unloading of stress-induced martensite, which can be safely deduced from the experimental (neutron diffraction) findings, showing a twinning mechanism in the unloading martensite regime. The reviewer is fully right that there is a slight rotation of the habit plane trace orientation with increasing strain (cf. Figs. 2b and c), which is an interesting and noteworthy detail. However, as habit plane motion/propagation is not relevant in the fully martensite regime and for the conclusions of the present work, further analysis of this phenomenon, either by experimental or theoretical work, is missing at this point. In light of the reviewer's remark, nonetheless, the authors have carefully revised the manuscript, clearly indicating the slight rotation of the habit plane trace orientation with increasing strain and the potential of theoretical works on this topic. (Page 14)

Questions concerning interpretation and discussion

19. Since no plastic strains is generated during the loading-unloading cycle, there is only marginal dislocation slip and it is unlikely the AE signal can be ascribed to it. The referee disagrees with the text on lines 420-425. Please provide a better explanation than dislocation slip in austenite. Dislocation slip in austenite in CoNiGa can be safely excluded at these stresses, particularly on unloading. If you do not know the source of this signal, just do not explain it.

RESPONSE #2.19: The authors thank the reviewer for this valuable remark and agree with the reviewer that the active $\langle 001 \rangle \{110\}$ slip system as well as the low stress levels applied upon loading and unloading of austenite should theoretically hamper any dislocation activities. Whether dislocation slip in the austenite has been de facto fully suppressed or a marginal amount of dislocation activities must be considered yet, cannot be finally assessed based on the results presented. In light of the reviewer's criticism, hence, not only the AE signals in the austenite regime are no longer attributed to dislocation slip (in the revised manuscript, the authors refer to future efforts to shed light on the origin of these signals), dislocation activities in general are no longer considered in this study as argued before. The authors have carefully revised the text as well as Fig. 7. (Pages 26, 27 and 29)

20. Figure 3 needs to show also habit plane trace.

RESPONSE #2.20: The authors thank the reviewer for this helpful suggestion. In the Co-Ni-Ga system, habit planes are of type $\{110\}$, i.e. of the same type as the twinning planes. Fig. 3 of the manuscript is specifically designed to clarify the CVP formation and corresponding twinning mechanism. In order not to overload the schematic, details about the habit planes have been added to the descriptions of Fig. 2, where these interfaces can be directly seen in the optical micrographs. Furthermore, additional information was also added to Fig. 7 (s. also response #2.22). (Pages 12 and 29)

21. The text on page 22 “After the forward transformation plateau, once CI 2 remains more or less constant, CI 1 shows a significant increase during both further loading in the martensite regime up to the maximum strain level of -5% (segment 4) ” is incorrect - the corresponding rate in Fig. 5c does not increase in segment 4 more than before.

RESPONSE #2.21: The authors thank the reviewer for this comment, however, must unfortunately disagree at this point. For clarity, an enlarged view of Fig. 6b of the manuscript, detailing the cumulated energy of cluster 1 in the time period from 30 – 70 s, is shown below in Fig. 2-1. As can be clearly seen from Fig. 2-1, there is an obvious increase of the rate from segment 3 to 4 and, thus, the statements in the original draft are considered to be correct.

Fig. 2-1: In situ AE results obtained during a SE compression cycle at 100 °C for an <001>-oriented Co-Ni-Ga single crystal in solution-annealed condition, showing the evolution of the cumulated AE energy of cluster 1 (red) in the time period of 30 – 70 s. For the corresponding evolution over the entire SE cycle (segments 1 – 7), please see Fig. 6b in the revised manuscript.

22. Since the load and face orientations of the crystal deformed in compression and volume fraction of twins (PTMC and experiment) are known, it is possible to plot exact traces of habit planes and twin planes on one of the sample faces (or even on a prism representing the sample). Such information needs to be included into figure 6

RESPONSE #2.22: The authors thank the reviewer for this valuable suggestion. In light of her/his remark, former Fig. 6 (Fig. 7 in the revised manuscript) has been revised by including more information about the habit and twin planes. (Page 29)

23. Page 25 lines 500-503 “ the authors claim that “ The excellent reversibility is governed by the easy motion of the stress-induced martensite plates of a single HPV system (all martensite plates have the same orientation with respect to the loading direction, cf. b, c, f). Since only fraction of surface on a single face was observed by OM it is possible that another variant appears elsewhere. The increasing stress in figure 2 beyond 2.5% strains suggest that possibility.

RESPONSE #2.23: As already mentioned in response #2.3, If only the OM results were available, the reviewer is right. In addition to the OM analysis, however, complementary neutron diffraction data are available and these data reveal the presence of just one CVP system that belongs to one single HPV system, i.e. martensite plates with the same orientation

with respect to the loading direction as seen in Fig. 2. In contrast to the surface information from the optical micrographs, the neutron diffraction results provide reliable structural information from the bulk volume. If a second internally twinned HPV system had formed during the SE loading-unloading cycles, then diffraction signals of a second CVP, i.e. a third (V_3) and fourth (V_4) martensite domain variant, should also have been detected on the eleven position-sensitive LiF/ZnS scintillator area detectors arranged around the sample position at SXD (ISIS neutron source, Rutherford Appleton Laboratory, Oxfordshire). However, this was not the case in the present study. For details on a martensitic morphology consisting of more than one single HPV system, the reviewer is referred to a previous study from some of the present authors [3], where such microstructure was found in $\langle 001 \rangle$ -oriented Co-Ni-Ga SMA single crystals by OM and neutron diffraction analysis after an artificial aging treatment. In light of the valuable comment by the reviewer and in order to substantiate that only a single HPV (CVP) system has formed in the present study, diffraction patterns with proper indexation following refinement is shown in the Supplementary Material (s. also responses #2.4 and #3.6).

24. Page 25 lines 503-506 “However, even though there is a lack of available slip systems in the $\langle 001 \rangle$ crystal direction under uniaxial loading, irreversible processes cannot be fully excluded, which in turn do not compromise the reversibility. Minor dislocation activities seem to appear in the B2 ordered parent phase over the entire SE cycle (a, b, f, g).” Dislocation slip creates slip traces and unrecovered plastic strain. The authors did not provide any evidence for dislocation slip in the experiment. Therefore, they cannot argue with it to interpret the experimental results.

RESPONSE #2.24: The authors thank the reviewer for this comment. The reviewer is right that no slip traces were found macroscopically on the sample surfaces, i.e. neither during the loading nor after the unloading path. And indeed, irreversible strain was also not detected after unloading. In light of the reviewer’s criticism already raised in her/his previous comment #2.19, hence, dislocation activities are no longer considered in this study. Whether dislocation slip in the austenite has been fully suppressed or a marginal amount of dislocation activities must be considered against theory needs further clarification in future work. For details on the theoretical framework and the revisions made by the authors in light of the reviewer’s comment, please see details in response #2.19.

25. Page 25 lines 510-513 “In the martensite regime, where the MT is assumed to be fully completed, twin re-orientation by twin boundary motion takes place. While upon loading detwinning by the growth of the dominant martensite domain variant at the expense of second, non-favorable oriented domain is assumed, ...” Although the activation of detwinning on forward loading is logical, no experimental evidence was provided for that.

RESPONSE #2.25: The reviewer is fully right with her/his comment that there is no experimental evidence for detwinning on the forward loading path. As already mentioned in the original draft on page 13 and now after careful revision according to response #1.1 to reviewer #1, only the “unloading regime of the martensite was selected for the neutron diffraction experiments due to the following two reasons: (1) Superimposed effects by a simultaneous reverse transformation of martensite into austenite were not to be expected and, thus, did not affect the evaluation of a twin re-orientation mechanism. (2) In comparison to the loading regime of martensite (s. segment 4 in Fig. 6b), the intensity of AE signals was found to be unexpectedly higher (s. segment 5 in Fig. 6b), suggesting more pronounced twin re-orientation activities in this region.” The detwinning during forward loading in the martensite regime is only assumed (as stated in the manuscript), but it is logical as the reviewer also confirmed. (Page 15)

Comments to the Conclusions

26. Page 26 lines 522-524 “The complementary findings obtained from the imaging, diffraction and acoustic measurements allowed a direct correlation between AE signals and the

deformation behavior, i.e. dislocation activities, stress-induced MPT and twinning processes.” This claim is not fully supported by the results (dislocations, only fraction of surface observed)

RESPONSE #2.26: The reviewer is right with his criticism. While the stress-induced MPT and twinning processes are clearly revealed by the optical microscopy and neutron diffraction analysis, respectively, direct experimental evidence for dislocation activities is not presented in the present study. It is only possible to infer these dislocation activities from the AE signals (s. comment #2.24). In light of the criticism, hence, the sentence has been carefully revised and now refers exclusively to the stress-induced MPT and twinning processes. (Page 31)

27. Page 26 lines 525-536 The text discusses low AE energy signals but it is probably more important to mention that the authors associate the low median frequency signal with detwinning process. The referee understands that low AE energy signal is low median frequency signal, yet still this needs to be mentioned in Conclusions

RESPONSE #2.27: The authors thank the reviewer for her/his valuable hint and have revised the Conclusions accordingly. (Page 31)

Reviewer #3: This particular work contains interesting results regarding acoustic emission caused in general by stress-induced martensitic transformation. It is an original one and shows the significance to the field, especially while it is related to methodology of the acoustic signals evaluation. Authors considered how microstructural features are evolving in the martensite state, which is important for shape memory alloys in general and for superelastic behavior in particular. In the text I left a number of comments and corrections that might be of help (all of it is in attached file - comments, inserts etc in pdf). Some required ones are related to some works published previously. Yet, these papers that are now missing in the text of the present paper are important but in general their absence do not prohibit publication. On the other hand, I'd like to insist on a proper corresponding revision. Also it can be good for the paper to include actual neutron diffraction patterns with proper indexing and showing exactly what are those variants that change their volume fraction upon unloading (that is also related to reproducibility issue). Some additional optical microscopy pictures somewhere between point e and f in Fig. 2 might also make obtained results more clear. My conclusion is that this paper will be ready for publication after revision.

RESPONSE #3: The authors appreciate the positive feedback of the reviewer and thank her/his for the valuable comments and helpful suggestions! The pure suggestions for alternative expressions that were raised by the reviewer in the attachment have been taken over in the revised version of the manuscript. All other remarks which need (more) clarification are replied in the following:

1. Page 3, ll. 47,48: This part of the sentence seems to be incomplete.

RESPONSE #3.1: The authors fully agree and have revised the sentence. (Page 3)

2. Page l. 57: It is not forbidden for the authors to put their own papers as reference. On the other hand, there is a requirement that important previous papers are taken into account. Here at least two of those that are missing. One, written by A. Planes, L. Manosa and E. Vives (<https://doi.org/10.1016/j.jallcom.2011.10.082>), was published in 2013 and represents some kind of a review for number of AE related papers. In there it is possible to find cases similar to the one described in the present paper under review. Moreover, even <001> single crystals are considered and strain- and stress-induced MPT as well. Second one, by the same group from Barcelona and their colleagues from Germany and Czech Republic (<https://doi.org/10.1103/PhysRevB.89.214118>) from 2014 shows a similar to the present paper approach, while using simultaneously AE and optical microscopy at temperature induced MPT in NiMnGa. In this way, it was possible to put into correspondence certain microstructural features and AE signals. My recommendation to authors is not only to include these papers in the reference list (it seems to be mandatory) but to use their results in a description of results and discussion section.

RESPONSE #3.2: In light of this valuable comment by the reviewer, the manuscript, i.e. not only the Introduction (Pages 4 and 5) but also the Discussions (Page 29) have been carefully revised. Now, these important previous papers are adequately referenced, which clearly helps to further improve the quality of the manuscript.

3. Page 5, l. 102: Does it mean that tubes were broken in air or into water?

RESPONSE #3.3: The authors apologize when the wording was unprecise at this point. The heat treatment procedure was the same as in three previous studies [2–4]. Following solution-annealing, the samples cooled down to room temperature at air conditions. For more clarity, the sentence has been revised. (Page 6)

4. Page 11, ll. 205, 206: This statement seems to be questionable. For the similar Heusler NiMnGa system there is a certain possibility to discern martensite domain variants by OM and clear models of magnetic shape memory are built upon this fact. How CoNiGa Heusler can be different from the above mentioned case?

RESPONSE #3.4: The authors thank the reviewer for this valuable hint. In the Co-Ni-Ga system, however, martensite domain variants were found in previous studies to be in the nanometer range (s. response #1.2 and Refs. [1,7]), which cannot be resolved by the optical microscopy setup used in the current study. In order to weaken the generalizing character of the statement, the sentence has been carefully revised. Now, the limitations of the present setup are marked down, which do not allow to observe the martensite domain variants (twins) in the current study. (Page 12 and s. also response #3.9 and #1.2)

5. Is there any OM pictures close to f but prior to reverse MPT like point 2 in Fig. 4? In other words, why not to put point f from Fig. 2 into correspondence with point 2 in Fig. 4? It seems logical. Is it there just one domain martensite variant according to OM below e (Fig. 2) down to point 2 in Fig. 4?

RESPONSE #3.5: The authors thank the reviewer for this comment and can fully understand her/his query about an OM image at point 2 in Fig. 4. Unfortunately, such OM image is not available. Nonetheless, the experimental results presented are clear evidence that indeed there is just one HPV system comprising a single CVP system upon unloading in the martensite regime, i.e. below point e in Fig. 2 up to the reverse MPT. First and foremost, the neutron diffraction results in Fig. 4 reveal the presence of just two martensite domain variants, i.e. a single internally twinned CVP system which corresponds with a single HPV system! In case of a multi HPV morphology, there would be at least four martensitic domain variants forming two CVP systems, as it was demonstrated in the presence of nanometric secondary phase particles in two previous studies [2,3]. Secondly, a single HPV configuration can also be deduced from the OM images in Fig. 2. If in point e of Fig. 2 a multi HPV morphology would be present, then traces of habit planes of a second orientation (at least!) with respect to the loading direction would appear with the start of the reverse MPT, which however is not the case in point f of Fig. 2, indicating a single HPV configuration in the unloading regime of martensite. And thirdly, the AE results clearly show the start of the reverse MPT upon unloading. Even though there is no OM image available before the reverse MPT sets in, the cumulated energy of CI 2 (blue curve in Fig. 6b of the revised manuscript) reveals MPT processes, starting when the critical stress for the reverse MPT is reached ($t \sim 73$ s), not before! For more clarity on this fact, the manuscript has been carefully revised. (Page 22)

Eventually, the combination of all in-situ results is clear evidence that no reverse MPT sets in between point 1 and 2 in Fig. 4, a single HPV system is present consisting of a single CVP system, and the AE signals detected in the unloading martensite regime (red curve in section 5 of Fig. 6b in the revised manuscript) can be attributed to a change in the volume ratio of the existent CVP, i.e. re-twinning.

6. Fig. 4, page 14: In addition to schematics in Fig. 4c and values of diffraction intensities in Fig. 4b, there is a need in corresponding neutron diffraction patterns. We believe authors but seeing the actual patterns with proper analysis like Rietveld with proper indexation of the observed peaks of interest (together with others) is much more valuable.

RESPONSE #3.6: The authors appreciate that the reviewer does not doubt findings presented, but the authors can also understand this criticism that was previously raised by reviewer #2 as well. To increase confidence in the results shown, now, details of the neutron diffraction data including diffraction patterns and proper indexation following refinement is presented in more detail in the Supplementary Material.

7. Page 19, ll. 378-381: Finally, there is an explanation for "elastic" regime. Still, I can not agree with the point that unelasticity is local. Look at the stress-strain macroscopic behavior upon unloading in Fig. 2 or Fig. 4a (the same dependence actually). The changing (!) slope upon unloading from point e in Fig. 2 says it all. This particular unelasticity is visible and can not be neglected especially because there is major microstructural effect behind it that authors are focusing upon in this paper.

RESPONSE #3.7: The authors thank the reviewer for this valuable comment. The authors fully agree with the argumentation raised by the reviewer and, thus, have revised the wording throughout the manuscript. Instead of "elastic martensite regime", in the revised version, it referred to as "unloading martensite regime". (Changes throughout the manuscript)

8. Page 23, l. 468: It can be also added that while authors of (<https://doi.org/10.1103/PhysRevB.78.094104>) while studying different single crystal system (CuZnAl) and in tension, paying most of their attention to the loading path associated with front propagation, still have shown in Fig. 1a that upon unloading there are AE signals prior to reverse MPT. There is a strong possibility of processes similar to ones observed and detailed in the present paper were taken place in CuZnAl.

RESPONSE #3.8: The authors appreciate the reviewer for this valuable hint and fully agree with her/his opinion. There seems to be a strong possibility that the AE signals observed by those authors in the Cu-Zn-Al system upon unloading of stress-induced martensite but prior to the reverse MPT might result from the same process as in the present study, i.e. twinning. In light of this valuable comment, now, the study indicated by the reviewer is not only referred to in the Discussion part (Page 29) but also in the Introduction (Page 4) of the revised manuscript. While those authors observed similar AE signals, the origin of those remained unclear. It has to be noted that Bonnot et al. recorded not the full AE data stream. They worked with threshold-based AE data recording. Moreover, they analyzed the AE signals only in the time domain counting for the AE activity (dN/dt). However, by this procedure the physical nature of the AE sources remains unclear. More evidence on the ongoing microstructural processes is obtained from the analysis of the AE data in the frequency domain. Nonetheless, that study on CuZnAl clearly strengthens the motivation for the current study and eventually shows its benefit for researchers working in this field. To the authors opinion, only the detailed analysis of the power spectral density function allows for a distinction between different microstructural processes acting as AE sources.

9. "Invisibility" of martensite domains in OM seems questionable. It could be the case that single domain martensite might have a different domain nucleation and growth at the expense of the existing one. It might be that these two martensite domain variants indeed exist in point e (Fig. 2). Still, it would be interesting to see evolution of OM pictures between point e (Fig. 2) to point 2 in Fig. 4 and below but prior to reverse MPT. Anybody there? Simple neutron diffraction intensities can not be considered as direct evidence of existing microstructure. The change in intensities might be. No one disputes the fact that there is a growth of one variant over another. Still, it is not known what the starting point (Fig. 2e) really looked like. The picture (Fig. 2e) shows single variant...

RESPONSE #3.9: As already stated in the previous response #3.4 to this reviewer, the authors fully agree with her/him that "invisibility" of martensite domain variants when OM is used does not apply in principle. In fact, it depends on the SMA system and the equipment used. For example, Bronstein et al. [8] clearly showed twinned martensite domain variants (CVP) using OM in single-crystalline NiMnGa. In the present case, however, such domain variants are not visible in the optical micrographs as a result of the equipment used and the size (thickness) of the martensite domain structures in the Co-Ni-Ga system [1,7] (s. also responses #1.2 and #3.4).

Furthermore, the reviewer is fully right that the picture in Fig. 2e shows a "single variant" configuration. However, it is of utmost importance to understand that this a single HPV

configuration. The single HPV system in Fig. 2e, in turn, is internally twinned and consists of two martensite domain variants, i.e. a single CVP system, as unequivocally shown by the neutron diffraction results in Fig. 4. For the query about the missing OM image in the unloading martensite regime before the reverse MPT sets in, please see response #3.5.

References

- [1] D. Niklasch, J. Dadda, H.J. Maier, I. Karaman, Magneto-microstructural coupling during stress-induced phase transformation in Co₄₉Ni₂₁Ga₃₀ ferromagnetic shape memory alloy single crystals, *J Mater Sci* 43 (2008) 6890–6901. <https://doi.org/10.1007/s10853-008-2997-z>.
- [2] C. Lauhoff, A. Reul, D. Langenkämper, P. Krooß, C. Somsen, M.J. Gutmann, I. Kireeva, Y.I. Chumlyakov, W.W. Schmahl, T. Niendorf, Effect of nanometric γ' -particles on the stress-induced martensitic transformation in $\langle 001 \rangle$ -oriented Co₄₉Ni₂₁Ga₃₀ shape memory alloy single crystals, *Scr. Mater.* 168 (2019) 42–46. <https://doi.org/10.1016/j.scriptamat.2019.04.003>.
- [3] C. Lauhoff, A. Reul, D. Langenkämper, P. Krooß, C. Somsen, M.J. Gutmann, B. Pedersen, I.V. Kireeva, Y.I. Chumlyakov, G. Eggeler, W.W. Schmahl, T. Niendorf, Effects of aging on the stress-induced martensitic transformation and cyclic superelastic properties in Co-Ni-Ga shape memory alloy single crystals under compression, *Acta Mater.* 226 (2022) 117623. <https://doi.org/10.1016/j.actamat.2022.117623>.
- [4] A. Reul, C. Lauhoff, P. Krooß, C. Somsen, D. Langenkämper, M.J. Gutmann, B. Pedersen, M. Hofmann, W.M. Gan, I. Kireeva, Y.I. Chumlyakov, G. Eggeler, T. Niendorf, W.W. Schmahl, On the impact of nanometric γ' precipitates on the tensile deformation of superelastic Co₄₉Ni₂₁Ga₃₀, *Acta Mater.* 230 (2022) 117835. <https://doi.org/10.1016/j.actamat.2022.117835>.
- [5] A. Weidner, A. Vinogradov, M. Vollmer, P. Krooß, M.J. Kriegel, V. Klemm, Y. Chumlyakov, T. Niendorf, H. Biermann, In situ characterization of the functional degradation of a [001⁻] orientated Fe–Mn–Al–Ni single crystal under compression using acoustic emission measurements, *Acta Mater.* 220 (2021) 117333. <https://doi.org/10.1016/j.actamat.2021.117333>.
- [6] L. Heller, P. Sittner, On the habit planes between elastically distorted austenite and martensite in NiTi, *Acta Materialia* 269 (2024) 119828. <https://doi.org/10.1016/j.actamat.2024.119828>.
- [7] J. Dadda, H. J-rgen Maier, I. Karaman, Y. Chumlyakov, High-temperature in-situ microscopy during stress-induced phase transformations in Co₄₉Ni₂₁Ga₃₀ shape memory alloy single crystals, *IJMR* 101 (2010) 1–11. <https://doi.org/10.3139/146.110427>.
- [8] E. Bronstein, E. Faran, D. Shilo, Analysis of austenite-martensite phase boundary and twinned microstructure in shape memory alloys: The role of twinning disconnections, *Acta Materialia* 164 (2019) 520–529. <https://doi.org/10.1016/j.actamat.2018.11.003>.

RESPONSE TO REVIEWERS' COMMENTS

We thank the reviewers for their positive feedback. The changes we made to address their valuable comments and useful suggestions are explained in our response to the reviewers and are marked **yellow** in the "marked up manuscript" version, respectively.

Reviewer #1: You responded to most of my comments in a satisfactory way. From my point of view, the paper can be published now.

Reviewer #3: Considering the fact that authors have taken into account most of my remarks and did their own discussion work in the vicinity of these remarks, I suggest that with all the present changes this paper can be published.

RESPONSE: The authors sincerely appreciate the recommendation for publication by Reviewer #1 and #3!

Reviewer #2: The authors responded to all questions of all 3 referees and made appropriate changes in the manuscript which resulted in improvement. However, I complained in my previous comments that "interpretation of the results in the manuscripts raises lots of questions and issues that need to be resolved before the manuscript can be recommended for publication" and this unfortunately did not really happen. Many issues still remain in the revised manuscript. Let me raise only two, which I consider to be most important
As concerns my questions concerning AE experimental results (4-14), the authors were not able to provide answers to most of them. Although that is understandable (results of AE experiments are very difficult to interpret), I am not sure if this is appropriate for a nature paper.

More problematic is the response of the authors to the questions 15-18 regarding the experimental evidence for twinning obtained by AE and ND. The referee is satisfied neither with the response in the rebuttal nor with the changes made in the manuscript. If tetragonal single crystal is loaded in compression, the martensite phase appearing behind the propagating habit plane is composed of martensite variants V1 and V2 providing same strain along the load axis (see Fig. 3). Because of that, there is no driving force for activation of detwinning on loading and twinning on unloading in the martensite state. If the authors have followed the referee suggestion in comment 20 "Figure 3 needs to show also habit plane trace", they would figure out this.

In spite of that, ND experiment detects small change of the peak intensities on unloading, which provides an evidence that detwinning occurred on unloading between the two measurement points. This is probably due to slight misalignment in the experiment, which is common in single crystal experiments. If the authors make similar experiment in tension, two martensite variants would provide very different strain to the load axis, there will be driving force for detwinning on loading and unloading in martensite state. They should have realized this simple consequence of deformation geometry and the experiment should have been performed in tension.

Second issue concerns the author's response to the comment related to the change of the habit plane trace orientation observed by OM upon loading and unloading. The authors made changes in the manuscript which acknowledge the experimentally observed changes of the habit plane trace. That is fine. But it seems they did not figure out the reason for it (or at least they did not explain it properly in the manuscript).

The authors deal with this on page 13 line 273-281. „The latter, in turn, is seen to be based on incompatibilities and further accommodation processes between the austenitic parent phase

and the twinned martensite to maintain a strain compatible habit plane interface. Theoretical work on this topic is available in literature, for example in Ref. [28], where the classical phenomenological theory of martensite crystallography (PTMC) [29] was extended to include the effect of elastic deformation of the austenite and martensite lattices due to external stress. However, for the discussions on the martensite variant selection, i.e. the number of HPV systems, these slight orientation differences between the habit plane traces can be neglected.“ However, the above text is rather confusing, as it mixes different phenomena and does not explain the experimental observation. Considering only MT (i.e. no dislocation slip), according to the PTMC theory, habit plane orientation is linked to the shape strain of the martensite. If shape strain of the martensite changes slightly by detwinning on loading and significantly by twinning on unloading, habit plane interface propagating during the reverse MT on unloading must differ (lie on a different austenite plane) from that which propagated on forward loading. Or there would have to be other mechanism assuring strain compatibility at the propagating habit plane interface interface (dislocation slip or additional twinning prior the habit plane in martensite) propagating during the reverse MT. If authors believe that PTMC theory is correct, they have to acknowledge this effect. Of course, the effect is small because the amount of twinning is small due to the absence of the driving force for it.

However, even if the effect is small, it is a consequence of a nature law (this referee believes that PTMC theory is right in the treatment of the requirement of strain compatibility at the habit plane interface as a nature law) and nature laws shall not be neglected in nature papers. The authors mention the effect of stress on habit plane orientation of stress induced MT elaborated in Ref. [28]. Yes, that is quite possible too, but this is a very different effect, which this referee did not have in mind while commenting on this. Classical PTMC theory neglecting stress effects on habit plane is sufficient to explain the observed habit plane changes.

Although the two comments raised above are rather negative, the referee is not negative to the whole manuscript. Appropriate modifications in the revised manuscript may solve these issues.

RESPONSE: The authors apologize that the questions within the first revision round were considered to be answered not satisfactorily. In order to address adequately now the remaining (most problematic) questions related to the martensitic transformation and the martensite reorientation mechanism (detwinning and re-twinning), we invited Prof. Hanuš Seiner from the Czech Academy of Sciences, a distinguished expert in the crystallographic theory of martensite, to contribute to this study (including a co-authorship for him). A mathematical model based on the phenomenological theory of martensite (PTMC) has been elaborated, describing the martensitic microstructure in the studied $\langle 001 \rangle$ -oriented Co-Ni-Ga single crystals under applied compressive loading. The microstructure of martensite that forms under such loading is seen to be (a close approximation of) an energy minimizer under three distinct types of boundary conditions:

1. the total axial contraction of the crystal being set by the grips of the loading device;
2. fixed zero in-plane strains and displacements in the contact area between the crystal and the grips, provided that the sample/crystal cannot freely glide along the grips (which is the case in the reported experiment);
3. compatibility conditions at the internal interfaces, in particular the habit planes, that dictate the volume fractions of the domain variants of martensite close to these interfaces.

Based on this boundary conditions, the evolution of the habit plane variant (HPV) microstructure as well as the correspondent variant pair (CVP) formation including martensite domain reorientation mechanisms have been established. For all details on these findings, the reviewer is referred to the changes made throughout the manuscript and, in particular, to the extensive discussions added to the *Supplementary Material*. As the numerical/theoretical results are fully in line with the experiments, the revisions clearly improve the quality of this study.

Finally, the authors would like to note the following points:

- In light of the new findings obtained in this second revision round, the authors have completely removed the sentence on former page 13 line 273-281.

- In the revised manuscript, the mechanisms “twinning/re-twinning” are referred to jointly as martensite reorientation mechanisms.
- The findings and considerations presented in the revised documents are valid for the present loading condition, i.e., compression along the $\langle 001 \rangle$ crystal direction. Tensile loading along $\langle 001 \rangle$, for example, is not accompanied by the formation of two martensite domain variants (CVP) as shown in our previous study (A. Reul et al., *Acta Mater.* 230 (2022) 117835, <https://doi.org/10.1016/j.actamat.2022.117835>). In contrast to compression, only a single martensite domain with its c-axis parallel to the loading direction forms, leading to a single domain, fully detwinned martensitic microstructure in tension. How acoustic emission signals appear under such microstructure scenario is to be investigated in future work.

RESPONSE TO REVIEWERS' COMMENTS

The authors sincerely appreciate recommendation for publication by all reviewers.

Reviewer #2: I am happy to recommend the revised manuscript for publication.

Below are comments to few bugs i spotted while reading the revised manuscript.

- Page 12/line 239 near (101) plane of the austenite
- Page 14/line 277 all habit planes that formed during forward and reverse transformation can be considered
- Page 17/line 347 there would be small variations of martensite lattice rotations

RESPONSE: The authors thank the reviewer for these comments. All suggestions were addressed in the revised manuscript.

Discussion with the Authors and suggestions for alternative expressions

Line no.	Present form	Suggested form (change)	Obs.
97-98	Following EDM, samples were mechanically ground to remove any residue from machining.	Following EDM, the surfaces of the samples were mechanically ground to remove any residue from machining.	Maybe only the investigation surface was ground
139-140	In situ neutron diffraction was carried out using the single crystal Laue diffractometer SXD [21] at ISIS neutron source ...		SXD needs to be explained, first.
227-229	Upon unloading from -5%, where the MPT is fully accomplished (Fig. 2e), the stress induced martensite becomes thermodynamically unstable and the reverse transformation to austenite sets in (Fig. 2f-h).	Upon unloading from -5%, where the MPT is fully accomplished (Fig. 2e), the stress induced martensite has been thermodynamically unstable and the reverse transformation to austenite sets in (Fig. 2f-h).	Stress-induced martensite has been unstable ever since it formed.
247-250	This elastic unloading regime ... was chosen ... since in this region both no superimposed effects by a simultaneous reverse transformation of martensite into austenite were to be expected (cf. Fig. 2) and, at the same time, the intensity of AE signals was found be unexpectedly high ...	This elastic unloading regime ... was chosen ... since in this region neither superimposed effects by a simultaneous reverse transformation of martensite into austenite were to be expected (cf. Fig. 2) nor the intensity of AE signals was found be unexpectedly high ...	
269-270	The integrated diffraction peak intensities calculated from data recorded at -5% and -4.5% applied compressive strain are depicted in Fig. 4a.	The integrated diffraction peak intensities calculated from data recorded at -5% and -4.5% applied compressive strain are depicted in Fig. 4b .	
295-296	In other words, the intensity ratio V_2/V_1 changes from 0.68 (-5%) to 0.73 (-4.5%), revealing twinning via twin boundary motion as schematically illustrated in Fig. 4c.		The present reviewer suggests that, if possible, AFM micrographs with corresponding surface profiles should be recorded in points (1) and (2)
371-372	In the present study, the deformation behavior of Heusler-type Co-Ni-Ga SMA single crystals have been assessed under SE compressive ...	In the present study, the deformation behavior of Heusler-type Co-Ni-Ga SMA single crystals has been assessed under SE compressive ...	
490-491	The solid and dotted circles depict microstructural features on the meso- and nanoscopic scale, respectively.		No dotted circles are noticeable in Fig.6!
508-511	Under the present loading conditions ... all martensite plates comprise of the same two martensite domain variants, In the martensite regime, where the MT is assumed to be fully completed ...	Under the present loading conditions ... all martensite plates consist in the same two martensite domain variants, In the martensite regime, where the MPT is assumed to be fully completed ...	

S Author: redacted Subject: Strikeout Date: 02/07/2025 15:40:06
 Authors main finding is related to unelastic behavior upon unloading (re-twinning mentioned below) and, therefore, word "elastic" can not be used in this case.

H Author: redacted Subject: Highlight Date: 02/07/2025 15:42:36
 The same reasoning as in previous comment. It might be prudent to use "unloading" instead of "elastic" in this case.

Submitted to Nature Communications in April 2025

 On the origin of acoustic emission signals upon elastic unloading of
 stress-induced martensite in Co-Ni-Ga shape memory alloy

 C. Lauhoff^a, A. Weidner^b, R. Lehnert^b, A. Reul^c, T. Pham^a, M.J. Gutmann^d,
 P. Krooß^a, W.W. Schmahl^c, H. Biermann^b, T. Niendorf^b

 9 ^a Institut für Werkstofftechnik, Universität Kassel, Mönchebergstr. 3, Kassel 34125,
 Germany

11 ^b Technische Universität Bergakademie Freiberg, Institute of Materials Engineering,
 Gustav-Zeuner-Straße 5, 09599 Freiberg, Germany

13 ^c Department of Earth and Environmental Sciences, Applied Crystallography, Ludwig-
 14 Maximilians-Universität, Theresienstr. 41, Munich 80333, Germany

15 ^d ISIS Facility, Rutherford Appleton Laboratory, Chilton, Didcot, Oxfordshire OX11 0QX,
 United Kingdom

 *corresponding author. email: lauhoff@uni-kassel.de; phone: +49 561 804-3976;

 The superelastic deformation behavior of Heusler-type Co-Ni-Ga shape memory alloy
 (SMA) single crystals is investigated employing complementary *in situ* techniques. Findings
 obtained by optical microscopy, neutron diffraction and acoustic emission (AE) provide deep
 insights into the microstructural events taking place during compressive loading conditions.
 In addition to the martensitic forward and reverse transformation, ~~elastic~~ unloading of the
 stress-induced martensite is found to be accompanied by the emission of acoustic signals. In
 the **elastic martensite regime**, neutron diffraction gives clear evidence for a change in the
 volume fractions of the martensite domain variants upon unloading, i.e. re-twinning of a
 correspondent variant pair (CVP) by the growth of one martensite domain variant at the

1. Introduction

Over the past decades, shape memory alloys (SMAs) have received significant
 attention by both industry and academia. Due to large reversible strains and their dissipative
 potential, this class of smart materials is highly attractive for designing efficient and compact
 solid-state actuator and damping devices, respectively [1–3]. Their unique functional
 properties, i.e. shape memory effect (SME) and superelasticity (SE), are based on a reversible
 martensitic phase transformation (MPT), i.e. a diffusionless, solid-state phase transition
 between a high-symmetry austenitic parent phase and a low-symmetry martensitic product
 phase [1,2]. The SE effect, which is in focus of the present work, represents a mechanical
 memory behavior. At temperatures above the austenite finish temperature (A_f), where the
 SMA is in a fully austenitic state, the material deforms ~~by~~ stress-induced MPT and possible
 detwinning of twinned martensite. Since the stress-induced martensite is unstable at
 temperatures above A_f , the reverse transformation back to austenite occurs upon unloading
 and, as a consequence, the deformation is fully recovered [1,2].

The successful incorporation of SMAs into envisaged applications depends on a
 thorough understanding of the MPT behavior. At this point, *in situ* characterization
 techniques are highly recommended to study the ongoing microstructural processes under
 load. In the past, acoustic emission (AE) has been already used for characterizing various
 microstructural phenomena in plenty of different kind of materials, e.g. ultrafine-grained
 copper [4], transformation-induced plasticity (TRIP) and twinning-induced plasticity
 (TWIP) steels [5,6], as well as SMA systems [7,8]. In particular, AE is one of the methods
 of choice for studying microstructural phenomena in SMAs, since among the plethora of
 *in situ* characterization techniques the AE method is one of the scarce real-time methods,

H Author: redacted Subject: Highlight Date: 02/07/2025 15:50:19
 this part of the sentence seems to be incomplete..

T Author: redacted Subject: Replace Text Date: 02/07/2025 15:49:13
 due to

H Author: redacted Subject: Highlight Date: 02/07/2025 15:44:36

It is not forbidden for the authors to put their own papers as reference. On the other hand, there is a requirement that important previous papers are taken into account. Here at least two of those that are missing. One, written by A. Planes, L. Manosa and E. Vives (<https://doi.org/10.1016/j.jallcom.2011.10.082>), was published in 2013 and represents some kind of a review for number of AE related papers. In there it is possible to find cases similar to the one described in the present paper under review. Moreover, even <001> single crystals are considered and strain- and stress-induced MPT as well. Second one, by the same group from Barcelona and their colleagues from Germany and Czech Republic (<https://doi.org/10.1103/PhysRevB.89.214118>) from 2014 shows a similar to the present paper approach, while using simultaneously AE and optical microscopy at temperature induced MPT in NiMnGa. In this way, it was possible to put into correspondence certain microstructural features and AE signals. My recommendation to authors is not only to include these papers in the reference list (it seems to be mandatory) but to use their results in a description of results and discussion section.

providing volume integrated information at high time resolution in the range of microseconds
[9]. However, AE is an indirect method and, thus, needs to be corroborated by further
characterization techniques, i.e. either by diffraction (neutron, X-ray, and electron) or
imaging (optical and electron microscopy) techniques. In addition, further work is still
needed to explore its full potential.

In one of the previous studies detailing the MPT behavior of a Fe-Mn-Al-Ni SMA, AE
signals were detected during unloading of stress-induced martensite, which couldn't be
unequivocally assigned by the authors [7]. The aim of the present study is to shed light on
those signals by focusing on the microstructural mechanisms, which take place in the elastic
martensite regime of SMAs ~~during unloading~~. The material of interest in this work are Co-
Ni-Ga single crystals with $\langle 001 \rangle$ crystal orientation. Single crystals were used instead of
polycrystalline material to allow for a systematic assessment of the operant microstructural
mechanisms. In Co-Ni-Ga alloys, which are characterized by a pronounced anisotropy of the
MPT behavior [10,11], polycrystals suffer premature intergranular cracking upon thermo-
mechanical loading due to incompatibilities at high-angle grain boundaries [12–14]. Single
crystals with the loading axis along $\langle 001 \rangle$, in turn, were selected because of the high slip
resistance in this orientation, allowing for excellent transformation recoverability and cyclic
functional stability under SE testing conditions [11,15–17]. As a result of the active
$\langle 001 \rangle \{110\}$ slip system in the B2-ordered austenite, the choice of the $\langle 001 \rangle$ orientation
minimizes the effect of dislocation activities on the deformation behavior [18]. Detailed
*in situ* analysis, i.e. AE as well as optical microscopy (OM) and neutron diffraction, were
carried out to analyze the deformation behavior in detail during compression SE experiments.
The results obtained with these complementary techniques reveal the deformation

mechanisms operant during SE testing and clearly unfold the origin of the AE signals. For
the first time, twin boundary motion causing the growth of one martensite domain variant at
the expense of the other one, known in the literature as detwinning/re-twinning of twinned
martensite, is detected for the elastic martensite regime.

**2. Experimental**

*2.1 Material*

Employing induction melted Co-Ni-Ga ingots with a nominal chemical composition of
49Co-21Ni-30Ga (in at.%), large single crystals with sizes up to several centimeters were
grown by Bridgeman technique under helium atmosphere. The specific alloy composition
has been designed for enhanced functional performance, i.e. a high degree of strain
recoverability [16]. Rectangular samples with dimensions of $3 \times 3 \times 6 \text{ mm}^3$ were electro-
discharge machined (EDM) from one of the bulk crystals such that their longer (i.e. loading)
axes were parallel to the $\langle 001 \rangle$ crystal direction of the austenitic phase, while normal vectors
of the lateral surfaces were parallel to $\langle 100 \rangle$ and $\langle 010 \rangle$. Following EDM, samples were
mechanically ground to remove any residue from machining. In order to obtain a single-
phase material state free of any secondary phases, which can significantly affect the
deformation behavior [15,19,20], all samples were initially solution-annealed. The solution-
annealing treatment was conducted at 1200 °C for 12 h in sealed quartz glass tubes under
argon atmosphere, followed by **breaking manually the quartz tubes at ambient conditions.**
Employing differential scanning calorimetry (DSC), the characteristic transformation
temperatures were determined. Single-crystalline samples with masses of $\approx 15 \text{ mg}$ were
prepared and investigated using a PerkinElmer DSC 8500. Fig. 1 shows a characteristic DSC

plot obtained at heating and cooling rates of 20 K/min. The solution-annealed Co-Ni-Ga is
characterized by martensite finish (M_f), martensite start (M_s), austenite start (A_s), and
austenite finish (A_f) temperatures of -10 °C, -6 °C, 11 °C, and 14 °C, respectively.

**Fig. 1** DSC curve of single-crystalline Co-Ni-Ga in solution-annealed condition. The
characteristic transformation temperatures upon heating (A_s and A_f) and cooling (M_s and
M_f) are marked.

2.2 In situ superelastic testing

Accompanied by various *in situ* techniques, quasi-static uniaxial compression tests
were carried out at 100 °C. As can be deduced from the DSC chart in Fig. 1, the selected test
temperature ensured a fully austenitic material state prior to testing of the solution-annealed
samples and allowed for direct comparison with data obtained in previous studies [15,20].
Each SE single cycle test was run in displacement control at a nominal strain rate of
$1 \times 10^{-3} \text{ s}^{-1}$ up to a maximum strain of -5% upon loading and a minimum load of -200 N for
unloading. The maximum strain level was chosen with respect to the theoretical
transformation strain of the $\langle 001 \rangle$ crystal direction under compressive loading [10].

martensite plates can be distinguished from the austenitic parent phase by the optical contrast.
Due to their direct and well-defined interface with the austenitic matrix, such plates are
termed habit plane variants (HPVs). When $\langle 001 \rangle$ -oriented Co-Ni-Ga is subjected to
compressive load, the martensite within the HPVs comprises two twin-related domain
variants [10,15], called correspondent variant pair (CVP, s. following paragraph for more
details). However, the martensite domain variants, which are separated by twin boundaries,
are not visible on the mesoscopic scale resolved by OM.

Various models, e.g. Bain, Kurdjumow-Sachs, Nishiyama-Wassermann, Pitsch, and
Greninger-Troiano, have been introduced in the past to describe the lattice relationships
between the austenitic parent and martensitic product phase in different alloy systems. In
case of Heusler-type Co-Ni-Ga, the Bain model allows to specify the martensite variant
selection under uniaxial loading conditions in a simplified but precise manner. As
schematically illustrated in Fig. 3, in theory, a total of three different tetragonal domains
(Bain-correspondent variants, BCV) can be formed during the cubic-to-tetragonal phase
transition. The extensional c-axes of these domains are parallel to the main cubic axis of the
parent B2-austenite. However, upon compressive loading along the $\langle 001 \rangle$ crystal direction,
i.e. the loading condition in the present study, the formation of the martensite domain with
its c-axis parallel to the loading direction (BCV_3) is energetically suppressed. Only the two
domains with their c-axes perpendicular to the loading direction (BCV_1 and BCV_2) will occur
and eventually form internally twinned CVP systems. Twinning planes are of type $\{110\}$
and, thus, two possible twin configurations exist along (110) and $(\bar{1}10)$ under uniaxial
compression along $\langle 001 \rangle$ [15].

H Author: redacted Subject: Highlight Date: 03/07/2025 10:53:42
 Is there any OM pictures close to f but prior to reverse MPT like point 2 in Fig. 4? In other words, why not to put point f from Fig. 2 into correspondence with point 2 in Fig. 4? It seems logical. Is it there just one domain martensite variant according to OM below e (Fig. 2) down to point 2 in Fig. 4?

Fig. 3 Schematic illustrating the Bain orientation relationship of the tetragonal martensite domain variant orientations to cubic austenite. Recompiled from [15]

Upon unloading from -5%, where the MPT is fully accomplished (Fig. 2e), the stress-induced martensite becomes thermodynamically unstable and the reverse transformation to austenite sets in (Fig. 2f-h). In contrast to the forward transformation, this is not accompanied by the formation of a dominant plate, but rather the simultaneous nucleation of numerous interfaces (Fig. 2f). However, all habit planes formed during forward and reverse transformation are parallel to each other, i.e. they feature the same orientation with respect to the loading direction, which is horizontal in Fig. 2. As HPVs with the same orientation in turn consist of the same CVP system [15,20], principally, the stress-induced MPT of the solution-annealed single-crystalline material in the present study is characterized by the formation of a single type of internally twinned martensite. This MPT behavior is in excellent agreement with *in situ* data on (001)-oriented Co-Ni-Ga single crystals in solution-annealed condition already tested under compression previously [15,20], and the presence of only one single CVP system is also confirmed by the neutron diffraction analysis detailed hereafter.

3.2 *In situ* neutron diffraction

A deformation related to a twin re-orientation mechanism within a CVP system under
compression, i.e. growth and shrinkage of one martensite domain variant at the expense of
the other one during loading and unloading, respectively, cannot be adequately assessed via
*in situ* optical analysis (s. Fig. 2). Therefore, *in situ* TOF neutron diffraction experiments
have been additionally conducted in the martensite regime upon unloading (s. Fig. 4a) in
order to shed light on the martensite domain variant selection and volume ratios of the
individual domains. This ~~elastic~~ unloading regime of the martensite was chosen for neutron
diffraction experiments since in this region both no superimposed effects by a simultaneous
reverse transformation of martensite into austenite were to be expected (cf. Fig. 2) and, at the
same time, the intensity of AE signals was found be unexpectedly high (s. segment 5 in
Fig. 5d as well as details in the following sections). In the past, neutron diffraction was
demonstrated to be a valuable method for an in-depth phase analysis, providing structural
information of bulk samples [15,19,24]. As shown in previous studies for various SMA
systems, phase fractions of martensite domain variants and, thus, elementary deformation
mechanisms like twinning could be assessed [25–28].

H Author: redacted Subject: Highlight Date: 03/07/2025 11:37:18
 In addition to schematics in Fig. 4c and values of diffraction intensities in Fig. 2b, there is a need in corresponding neutron diffraction patterns. We believe authors but seeing the actual patterns with proper analysis like Rietveld with proper indexation of the observed peaks of interest (together with others) is much more valuable.

**Fig. 4** In situ neutron diffraction results obtained during a SE compression cycle at
 100 °C for an $\langle 001 \rangle$ -oriented Co-Ni-Ga single crystal in solution-annealed condition: (a)
 characteristic SE stress-strain curve. (b) diffraction intensities obtained for austenite (black
 triangles) as well as martensite domain variant V₁ (blue squares) and V₂ (green circles) at
 263 -5% (filled symbols) and -4.5% (open symbols), respectively, and (c) schematic illustrating
 the re-twinning process in the elastic martensite regime upon unloading. The specific stress-
 strain stages investigated are highlighted by the red point (-5%) and circle (-4.5%) in (a).
 The inset in (b) reveals the intensity variance (s^2) of the analyzed diffraction data. See main
 text for details.

The integrated diffraction peak intensities calculated from data recorded at -5% and
 270 -4.5% applied compressive strain are depicted in Fig. 4a. In general, such diffraction peak
 intensities correspond with the volume fraction of individual phases. Diffraction data at both
 stress-strain stages (cf. marks in Fig. 4a) revealed the presence of two martensite domain
 variants labeled as V₁ and V₂ in the remainder of the text, respectively. It is important to note
 that intensities related to additional domain variants were not detected. Only some minor
 intensities from the austenitic parent phase were found, indicating that the stress-induced

Author: redacted Subject: Replace Text Date: 03/07/2025 11:45:29
unloading - reasoning as above

Author: redacted Subject: Highlight Date: 03/07/2025 11:53:23

Finally, there is an explanation for "elastic" regime. Still, I can not agree with the point that unelasticity is local. Look at the stress-strain macroscopic behavior upon unloading in Fig. 2 or Fig. 4a (the same dependence actually). The changing (!) slope upon unloading from point e in Fig. 2 says it all. This particular unelasticity is visible and can not be neglected especially because there is major microstructural effect behind it that authors are focusing upon in this paper.

f_m of about 20 to 60 kHz. The energies E of this cluster are relatively low, however, still
above of those of the noise signals. It is well known from literature and previous studies of
some of the co-authors that low energy and low median frequency AE signals are related to
microstructural processes occurring with lower velocities such as dislocation movement or
the movement of interface phase boundaries [6]. In contrast, cluster 2 (Cl 2; blue signals)
contains signals with significantly higher AE energies E , which are also spread over a wider
range of median frequencies f_m from 20 up to 350 kHz. This kind of signals is known to be
related to fast processes occurring with the velocity of sound such as brittle crack formation,
MPT, mechanical twinning or even detwinning in hcp materials such as magnesium
[6,32,33].

4. Discussion

In the present study, the deformation behavior of Heusler-type Co-Ni-Ga SMA single
crystals have been assessed under SE compressive loading conditions employing various
complementary *in situ* techniques. Beside the OM observations showing the stress-induced
forward and reverse MPT accompanied by the movement of the interphase boundaries
between austenite and martensite (habit planes, cf. Fig. 2), the neutron diffraction data
obtained in the ~~elastic~~ martensite regime have clearly revealed a twinning mechanism, i.e.
the growth of the minor dominant domain variant (V_2) at the expense of the major one (V_1)
(cf. Fig. 4c) by the movement of twin boundaries. **In consequence, actually, the material**
**response in the martensite regime is not ideal elastic from a very local point of view. From a**
**global point of view, however, the material behavior in the martensite regime is seen as quasi**
**elastic** and, thus, was referred to as "elastic" for ease of naming. The main objective of the

of the stress plateau of the reverse transformation (segment 5). Here, the movement of twin
boundaries between V_1 and V_2 are supposed to contribute to signals of Cl 1, which can be
rationalized from the findings obtained by the neutron diffraction experiments (Fig. 4). The
martensite domain variant V_1 has the highest diffraction intensity at the maximum stress.
Upon unloading, the intensity ratio V_2/V_1 increases from 0.68 (-5%) to 0.73 (-4.5%). This
can be interpreted as the growth of V_2 at the expense of V_1 during unloading, known as re-
twinning. The twin boundary motion within the internally twinned CVP system causes the
steady increase in the cumulated AE energy of Cl 1 in the stage of unloading of the stress-
induced martensite. Even though not directly shown in the present study, equally, the
contribution of twin boundary motion is assumed to be responsible for the corresponding
signals of Cl 1 during the previous loading step up to the maximum stress level at -5% applied
strain. In this region, however, the opposite behavior is expected, i.e. detwinning, where V_1
is growing at the expense of V_2 . This behavior is known from one of the previous studies
[22], where detwinning of martensite have been observed in the loading regime, i.e. the
martensite domain variant, which is favorably oriented with respect to the loading axis grew
at the expense of the second, non-favorable oriented domain. The different slopes of Cl 1 in
the loading (segment 4) and unloading (segment 5) martensite regime, however, cannot be
explained at this point, and also requires additional work.

Finally, the reverse transformation into austenite sets in when the critical stress for its
onset is reached upon further unloading (cf. Fig. 2). Over the entire reverse transformation
plateau, then, the transformation into austenite not only leads to a steady accumulation of AE
energy belonging to Cl 2 (cf. Fig. 5d), it is also accompanied by a steady increase in the
cumulated AE energy of Cl 1 (segment 6). Noteworthy, the cumulated AE energy curve of

Based on the findings obtained within the current study, Fig. 6 schematically highlights
the deformation mechanisms, which has to considered for $\langle 001 \rangle$ -oriented Heusler-type Co-
Ni-Ga SMA single crystals in solution-annealed condition under SE compressive loading at
100 °C. The schematic is designed to allow for rapid access to the main findings of the
present work, which were already presented and discussed above. The solution-annealed and
secondary phase free material state is characterized by a SE behavior with full strain
recovery. This outstanding functional performance is based on the thermoelastic stress-
induced phase transformation from the B2-ordered austenitic matrix to a non-modulated
tetragonal martensite with $L1_0$ structure (b, c) and vice versa (f). The excellent reversibility
is governed by the easy motion of the stress-induced martensite plates of a single HPV system
(all martensite plates have the same orientation with respect to the loading direction,
cf. b, c, f). However, even though there is a lack of available slip systems in the $\langle 001 \rangle$ crystal
direction under uniaxial loading, irreversible processes cannot be fully excluded, which in
turn do not compromise the reversibility. Minor dislocation activities seem to appear in the
B2 ordered parent phase over the entire SE cycle (a, b, f, g). Beside the stress-induced MT
and the marginal amount of dislocation slip, finally, twinning has been identified as another
deformation mechanism. Under the present loading conditions, i.e. compression along $\langle 001 \rangle$,
all martensite plates comprise of the same two martensite domain variants, i.e. a single
internally-twinned CVP system. In the martensite regime, where the MT is assumed to be
fully completed, twin re-orientation by twin boundary motion takes place. While upon
loading detwinning by the growth of the dominant martensite domain variant at the expense
of second, non-favorable oriented domain is assumed, the opposite re-twinning process, i.e.
the growth of second, minor dominant domain (d, e), occurs during unloading.

H Author: redacted Subject: Highlight Date: 03/07/2025 12:23:57

"Invisibility" of martensite domains in OM seems questionable. It could be the case that single domain martensite might have a different domain nucleation and growth at the expense of the existing one. It might be that these two martensite domain variants indeed exist in point e (Fig. 2). Still, it would be interesting to see evolution of OM pictures between point e (Fig. 2) to point 2 in Fig. 4 and below but prior to reverse MPT. Anybody there? Simple neutron diffraction intensities can not be considered as direct evidence of existing microstructure. The change in intensities might be. No one disputes the fact that there is a growth of one variant over another. Still, it is not known what the starting point (Fig. 2e) really looked like. The picture (Fig. 2e) shows single variant...